# Single-step retrosynthesis prediction by leveraging commonly preserved substructures

**Lei Fang** [1] ✉, **Junren Li** [2], **Ming Zhao** [3], **Li Tan**[4] **& Jian-Guang Lou**[1]

Retrosynthesis analysis is an important task in organic chemistry with numerous industrial applications. Previously, machine learning approaches employing natural language processing techniques achieved promising results in this task by first representing reactant molecules as strings and subsequently predicting reactant molecules using text generation or machine translation models. Chemists cannot readily derive useful insights from traditional approaches that rely largely on atom-level decoding in the string representations, because human experts tend to interpret reactions by analyzing substructures that comprise a molecule. It is well-established that some substructures are stable and remain unchanged in reactions. In this paper, we developed a substructure-level decoding model, where commonly preserved portions of product molecules were automatically extracted with a fully data-driven approach. Our model achieves improvement over previously reported models, and we demonstrate that its performance can be boosted further by enhancing the accuracy of these substructures. Analyzing substructures extracted from our machine learning model can provide human experts with additional insights to assist decision-making in retrosynthesis analysis.

Organic synthesis is an essential branch of synthetic chemistry that mainly involves the construction of organic molecules through various organic reactions. Retrosynthesis analysis[1] that aims to propose possible reaction precursors given a desirable product is a crucial task in computer-aided organic synthesis. Accurate predictions of reactants can assist in finding optimized reaction pathways from numerous possible reactions. In the context of our paper, we use the term "reactants" to refer to substrates that contribute atoms to a product molecule. Solvents or catalysts that take part in the reaction, but do not contribute any atoms to the product were not considered as reactants in the context of our paper. Recently, machine learning-based approaches have achieved promising results on this task. Many of these methods employ encoder-decoder frameworks, where the encoder part encodes the molecular sequence or graph as high dimensional vectors[2–8], and the decoder takes the output from the encoder and generates the output sequence token by token autoregressively. The sequences of the molecules involved in these algorithms are usually represented as SMILES (Simplified Molecular-Input Line-Entry System) strings[9,10], and the graph refers to the molecular graph structure. For example, Molecular Transformer[2] and Augmented Transformer[3] used textual SMILES representations of reactants and products. Subsequently, retrosynthesis analysis was formulated as a machine translation task from one language (product) to another (reactants). Molecular Transformer[2] was applied to predict retrosynthetic pathways with exploration strategies based on Bayesian-like probabilities[11].

Casting retrosynthesis analysis as a machine translation task enables the use of deep neural architectures that are well-developed in natural language processing. For example, self-attention based Transformer architectures[12] are employed in recent state-of-the-art

[1]Microsoft Research Asia, No.5 Dan Ling Street, Beijing, China. [2]College of Chemistry and Molecular Engineering, Peking University, No.5 Yiheyuan Road, Beijing, China. [3]IPS, Waseda University, 2-7 Hibikino, Wakamatsu-ku, Kitakyushu-shi, Fukuoka 808-0135, Japan. [4]Mincui Therapeutix, No.1 Yongtaizhuang North Road, Beijing, China. ✉e-mail: leifa@microsoft.com

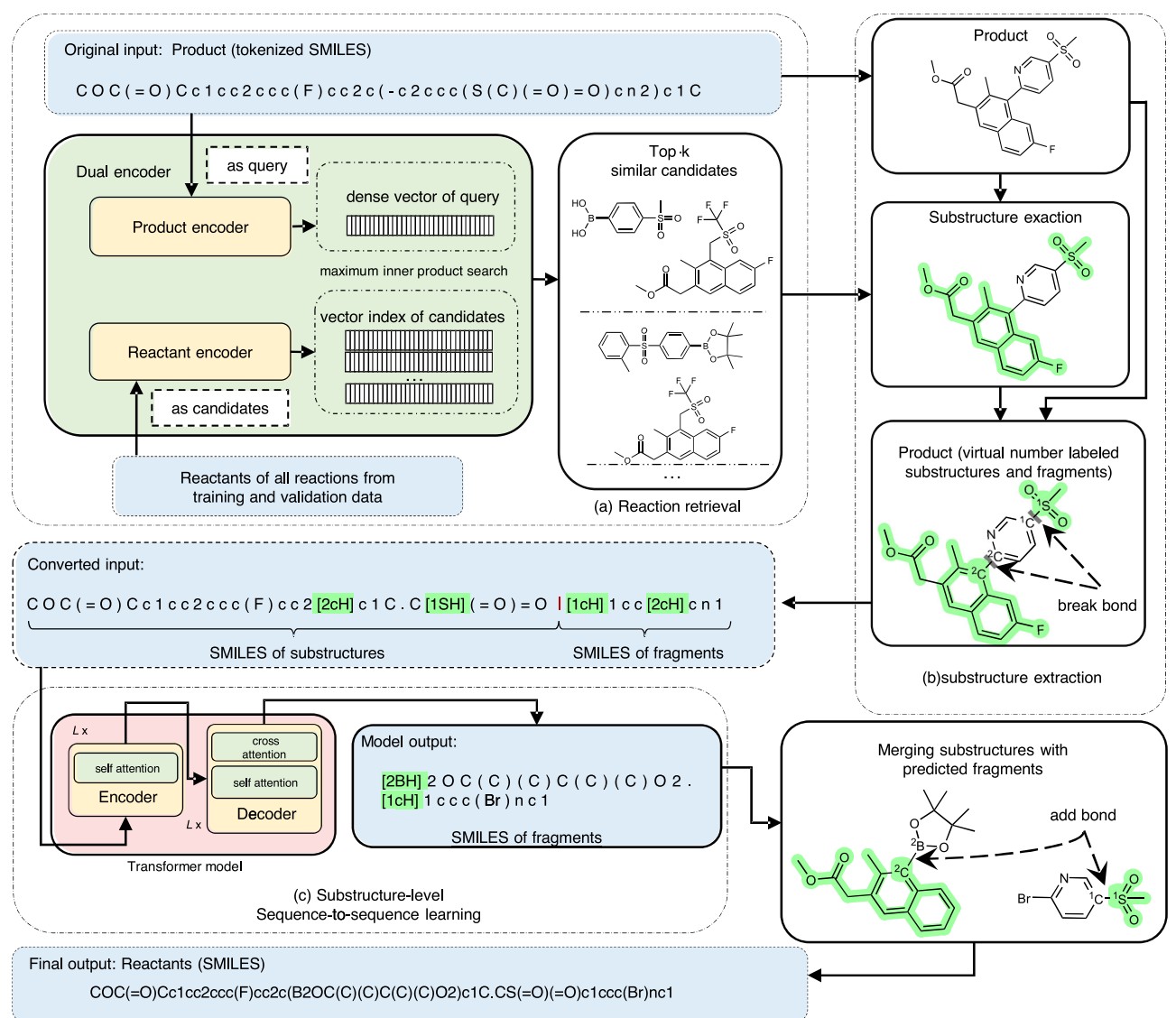

**Fig. 1 | Method overview, virtual number labeled atoms and substructures are highlighted in green. a** Reaction retrieval using the dual encoder model. **b** Substructure extraction and isolation using virtual numbers as labels. **c** Substructure-level sequence-to-sequence learning with Transformer encoder and decoder, both of which have a stack of $L$ identical blocks. The predicted fragments are merged with the substructures to obtain the final output. It is worth noting that the predicted fragment remains in the assembled reactant molecules, which is slightly divergent from chemical intuitions.

models[2–7]. In the decoding stage, output SMILES strings are auto-regressively generated token-by-token, where the elementary tokens in the SMILES strings from traditional approaches mostly involved individual atoms in a molecule. This is not immediately intuitive or explainable for chemists in synthesis design or retrosynthesis analysis. In real world route scouting tasks, synthetic chemists generally rely on their professional experience to formulate a reaction pathway by drawing inspirations from previously learned reaction pathways, coupled with an abstract understanding of the underlying mechanisms based on first principles. For human experts, retrosynthesis analysis often starts from molecular substructures or fragments that are chemically similar to, or are preserved in target molecules. These substructures or fragments help provide clues to an assembly puzzle involving a series of chemical reactions that may yield the final product.

In this paper, we propose to leverage commonly preserved substructures in organic synthesis, where the substructures extracted from large sets of known reactions capture subtle similarities among reactants and products, while remaining free from expert systems or template libraries. This way, we essentially cast the retrosynthesis analysis as a substructure-level sequence-to-sequence learning task. The pipeline of the overall framework is illustrated in Fig. 1, which consists of the following modules:

(a) *Reaction retrieval.* The reaction retrieval module aims to retrieve similar reactions given an individual product molecule as a query, and these retrieved associated reactions are extracted for commonly preserved substructures. We introduce a learnable cross-lingual memory retriever[13] used in machine translation tasks to align the reactants and the corresponding products in high dimensional vector space. The retrieval model is based on the dual-encoder framework[14]. For each reaction, the learned representation of reactants is similar to that of the product. After the dual-encoder retrieval model is trained, we obtain dense vector representations of all the reactants and products, as shown in Fig. 1a. In retrosynthesis analysis, the product representation forms the query to retrieve reactant molecules that are similar in high dimensional vector space. To conduct a fair comparison to other methods, the retrieved candidate reactants (denoted as

"candidates" subsequently) are only derived from the same training and validation data set utilized by previous studies.

(b) *Substructure extraction.* Given the training objective of the dual-encoder retrieval model, the retrieved molecules are expected to be similar to the correct, or "golden" reactants. We then extract the common substructures from the product molecule and the top cross-aligned candidates based on molecular fingerprints. We assume these common substructures to also exist in the golden reactants, and likely remain intact for the reactions considered. More details are provided in the section "Substructure extraction". The common substructures provide a reaction-level, fragment-to-fragment mapping between reactants and products. These substructures are product molecule specific and are analogous to reaction templates learned from the dual encoder retrieval model, rather than from an expert system. We then separate the molecules into common substructures and other molecular fragments. We use "molecular fragment(s)", or simply "fragment(s)" in the context of this paper to refer to those atoms and bonds not present in the common substructure. When multiple bonds are broken to isolate the substructures, we introduce "virtual number(s)" to virtually tag the atoms connected by the broken bonds, as shown in Fig. 1(b).

(c) *Substructure-level Sequence-to-sequence Learning.* With commonly preserved substructures and molecular fragments, we convert the original token-level sequence based largely on atoms, to a substructure-level sequence. The new input sequence is the SMILES strings of the substructures followed by the SMILES strings of other fragments with virtual number labels. The output sequences are the fragments with virtual numbers. In other words, the fragments are connected to common structures with bonds specified by these virtual numbers. Subsequently, retrosynthesis analysis is cast into a structure-level sequence-to-sequence learning task. Given the model predicted virtually labeled fragments to various locations on the substructures, we ultimately perform a bottom-up modular assembly of these individual pieces to obtain the final molecular graph and its SMILES strings. An example is shown in Fig. 1(c), where 1S (denoted by [1SH]) is a virtually labeled atom from the substructure that should be connected to atom 1c (denoted by [1cH]) in the predicted fragment. Similarly, 2c (denoted by [2cH]) from the substructure should be connected to atom [2B] (denoted by [2BH]) in the predicted fragment.

Substructure analysis is integral to how human researchers perform retrosynthesis analysis, and our approach achieves improvement over previously reported models. We demonstrate that the performance can be boosted further if we enhance the accuracy of substructure extraction. Substructures extracted from our model can potentially provide human experts additional insights for decision-making in routine synthesis tasks. While still early in development, we demonstrate that it's possible to develop a machine-learning model by mimicking human experts' way of thinking.

The remainder of this paper is organized as follows: in the section "Results", we present simulation results on retrosynthesis prediction and substructure extraction based on the model we developed. In the section "Discussion", we discuss and summarize the strengths and weaknesses of our approach when compared to existing models. In the section "Methods", we provide more details of how we built our model, including reaction retrieval, substructure extraction, and substructure-level sequence-to-sequence learning.

## Results

### Retrosynthesis prediction results
We report the overall results of one-step retrosynthesis based on the USPTO_full dataset in Table 1. For comparison, we analyzed the results

**Table 1 | The results of retrosynthesis on USPTO_full dataset**

| Models | Top-1 (%) | Top-10 (%) | *Templ.* | *Map.* |
|---|---|---|---|---|
| RetroSim [15] | 32.8 | 56.1 | yes | yes |
| MEGAN [16] | 33.6 | 63.9 | no | yes |
| GLN [17] | 39.3 | 63.7 | yes | yes |
| Graph2SMILES [8] | 45.7 | 63.4 | no | no |
| Augmented Transformer [3] | 44.4 | 70.4 | no | no |
| Ours | 46.0 | 68.5 | no | no |
| Ours[a] | 48.2 | 69.9 | | |
| RetroPrime [5][b] | 44.1 | 68.5 | no | yes |
| Augmented Transformer [3][b] | 46.2 | 73.3 | no | no |
| GTA [6][b] | 46.6 | 70.4 | no | yes |
| Ours[b] | 48.2 | 71.6 | | |
| Ours[ab] | 50.4 | 73.1 | no | no |
| Ours[c] | 50.2 | 73.8 | | |
| Ours[a,c] | 53.6 | 76.7 | | |

*Templ.* reaction templates used, *Map.* atom-mapping information used.
[a]Substructures are all correct.
[b]Invalid reactions are excluded from the test set.
[c]Results on data with substructures.

from other notable works in this area that employed different machine learning pipelines. RetroSim[15] treated retrosynthesis as a template ranking problem based on molecular similarity. MEGAN[16] morphed the problem into sequences subjected to molecular graph edits. GLN[17] employed a conditional graph logic network to learn chemical templates for retrosynthesis analysis. RetroPrime[5] decomposed a given product molecule into synthons and generated reactants through attachment of leaving groups. Augmented Transformer[3] incorporated data augmentation strategies with a base Transformer model. Graph2SMILES[8] combined a Transformer decoder with permutation invariant molecular graph encoders. GTA[6] proposed a molecular graph-aware attention mask for both self-attention and cross-attention when applying Transformer models.

The following characteristics of the USPTO_full data set are worth noting for future reference. On the test set, about 4.4% of products have no reactants, rendering them as invalid data; for the remaining products, approximately 82.2% successfully produced a substructure from our current pipeline. Not every product molecule is guaranteed to produce a substructure in our current implementation as the extraction process relied on comparisons between products and candidate reactants. For a fair comparison, we trained a vanilla Transformer model with augmented random SMILES to obtain predictions for products with no substructures. Substructures are considered "correct" if they are part of the golden reactants, and "incorrect" otherwise. We also attempted filtering out incorrect substructures and assessed the model based on the subset containing only correct substructures. As the improvement of our approach is dependent on successfully extracting substructures from product molecules, we present the results on this subset as well.

In almost every scenario, our method achieved comparable or better top 1 accuracy when compared to other methods previously tested. On the subset of data where substructures were successfully extracted, model performance is much improved compared to the overall result, as shown in Table 1. Because we employed a pair-wise ranker in our model to rerank predictions, the results in different scenarios were compared to those from a competing ranking strategy in the Augmented Transformer[3] model in Supplementary Results. Filtering out incorrect substructures further improved top *k* accuracies (see Table 1). This improvement demonstrates that performance metrics on the model can be further improved if we invest additional effort in enhancing the accuracy of substructure extraction, as substructures are essential to every aspect of our implementation and

forms the fundamental basis to simplifying molecules with human-like intuitions. Once again, it's worth noting that our approach does not require any reaction templates built upon expert systems or template libraries containing prior knowledge about organic chemistry, and does not consider any atom mapping information from reactants to products inherent in the data set. Atom-mapping information can reveal information about potential reaction sites[5].

The improvement in our method can be attributed to two main factors: 1) our method managed to successfully extract substructures from 82.2% of all products on the USPTO_full test data set, which is a relatively high coverage showing the general applicability of this approach, 2) we only needed to generate fragments connected to virtually labeled atoms in the substructures, which shortened the string representations of molecules, significantly lowering the number of atoms to be predicted. For product molecules with extracted substructures, the average number of atoms in reactants to be predicted is reduced from 30.0 to 17.9. Our model design has advantages over previous token-by-token decoding models, where the elementary tokens were mostly atoms, e.g., Augmented Transformer[3], Graph2SMILES[8] and GTA[6].

### Results on commonly preserved substructures

In the section "Substructure exaction", we describe in detail how substructures are obtained based on common fingerprints between a given product and the retrieved candidates. Because the retrieved candidates are not always golden reactants, errors can be introduced in the extraction process, resulting in incorrect substructures. For example, the substructure extracted from candidate #1 in Fig. 2 is incorrect. In this case, the retrosynthesis product is a long molecule linked by a triple bond. All retrieved candidates have a shared common substructure with the product. By taking a further look at the products associated with these candidates, we readily observe that the triple bond itself is likely to be the reaction site. This means that the triple bond should not have been included in the substructure, even if it is in the environment of the aligned fingerprint. We leave this as planned future work to improve the accuracy of substructure extraction, i.e., we plan to identify possible reaction sites based on the retrieved candidates and exclude these atoms from being considered as part of a substructure.

For incorrect substructures, we can easily filter them out with golden reactants on the training and validation data. On the training data set, we extracted substructures from 81.9% of product molecules after filtering incorrect substructures. The extracted substructures were derived from the complete set with 20 retrieved candidates. The average number of candidates that yielded a substructure is 12.5. The average number of unique substructures is 4.2. The model training data is formed by unique substructures only to avoid redundancy.

On the test data, we extracted substructures from 82.2% of product molecules with an accuracy of 90.2%. The average number of substructures and unique substructures are 12.1 and 4.9, respectively. The average number of heavy atoms in the product, substructures, and golden reactants are 26.3, 12.1, and 30.0, respectively. On the entire test data set of product molecules and their associated candidates, 79.8% yield at least one correct substructure, 63.0% yield structures that are all correct, and 2.4% (82.2−79.8%) yield all incorrect substructures. It is worth noting that if a particular substructure is incorrect, predictions based on that particular substructure are also incorrect.

To improve the accuracy of extracted substructures, we could identify potential reaction sites based on the retrieved candidates. Another viable method is to increase the threshold for common fingerprint selection from the retrieved candidates. Note that our current implementation requires that the common fingerprints exist in at least 5 out of the 20 retrieved candidates to define a successful substructure extraction. Fig. 3 shows the percentage of products with substructures,

the percentage of products with all correct substructures, and the accuracy of substructures, when the threshold is set between 3 and 10. The result shows that as accuracy increases, the likelihood of obtaining products with substructures monotonically decreases, while the percentage of products with all correct substructures forms a convex curve with a maximum at around 6. In this paper, we set the threshold to 5 mainly because this setting balances a relatively high percentage of products with successfully extracted substructures and a high percentage of these substructures being correct. On the test data, even if products and associated candidates yield results that are inclusive of incorrect substructures, the average number of correct substructures is still 7.3. This indicates that, after aggregating the results, the model can predict correctly even if substructure extraction resulted in a partially incorrect set of substructures.

Our hypothesis is that substructures are relatively stable and tend to remain unchanged during reactions, which can be associated with a low likelihood for inclusion of a reaction site. We discovered at least two distinct types of extracted substructures from a cursory analysis: (1) those located at the ends of the molecules with protections from other reactive functional groups in the same molecule. For instance, a hydroxy group protected by a trimethylsilyl group is a common substructure across different reaction types; (2) those located in the middle of the molecules with inert alkyl chains or aromatic rings, which contain no reactive functional groups. The percentages of substructures with aromaticity in the top-10 and 20 most frequent substructures are 80% and 70%, respectively. On average, among all the substructures, about 60% of the atoms are included in those aromatic ring structures. These numbers show that extracted substructures have some level of chemical interpretability even with this initial implementation of our model, which has minimal manual input of chemistry knowledge.

It is important to note that the extracted substructure is product molecule specific, which can help to capture subtle structural changes from reactants to products that are reaction specific. Phthalimide is a common heterocyclic substructure. We show four exemplary reactions, where the reactants contain phthalimide in Fig. 4. The extracted substructures vary among different reaction types. In the model output, phthalimide is not considered to be part of the substructure for reaction (a) and reaction (b). The substructures of reaction (c) and reaction (d) are different, yet they both contain phthalimide. The results show that substructures are product-specific, which is consistent with our expectations.

Another benefit of leveraging commonly preserved substructures is that we can provide users with additional insights for decision-making in retrosynthesis planning when compared with existing methods. For the case shown in Fig. 2, the product can be synthesized via multiple types of coupling reactions. Because we can group predictions by substructure, our predicted groups of reactants and reactions can aid human experts by giving them an opportunity to assess potential pathways, and eliminate infeasible reactions through chemistry knowledge. As is shown in Fig. 2, the reaction associated with the first candidate reactant is a Suzuki-Miyaura coupling reaction between benzene and thiophene rings, while the reactions associated with the remaining candidates are Sonogashira coupling reactions, with the triple bond being the reaction site. This example indicates that an expert user can refine the predictions by comparing reactions associated with retrieved candidates, making our predictions more explainable and trustworthy compared with existing "black-box" models.

Note that in Fig. 2, the extracted substructure is not a fully connected graph, as they come from different parts of the same molecule. As discussed in the section "Substructure exaction", broken bonds do not necessarily mean reaction sites. For instance, the broken bonds associated with substructures from candidate reactants #2, #3 and #4 in Fig. 2 are not reaction sites. While substructures produced by our current model are not indicative of actual chemical reactions, and our

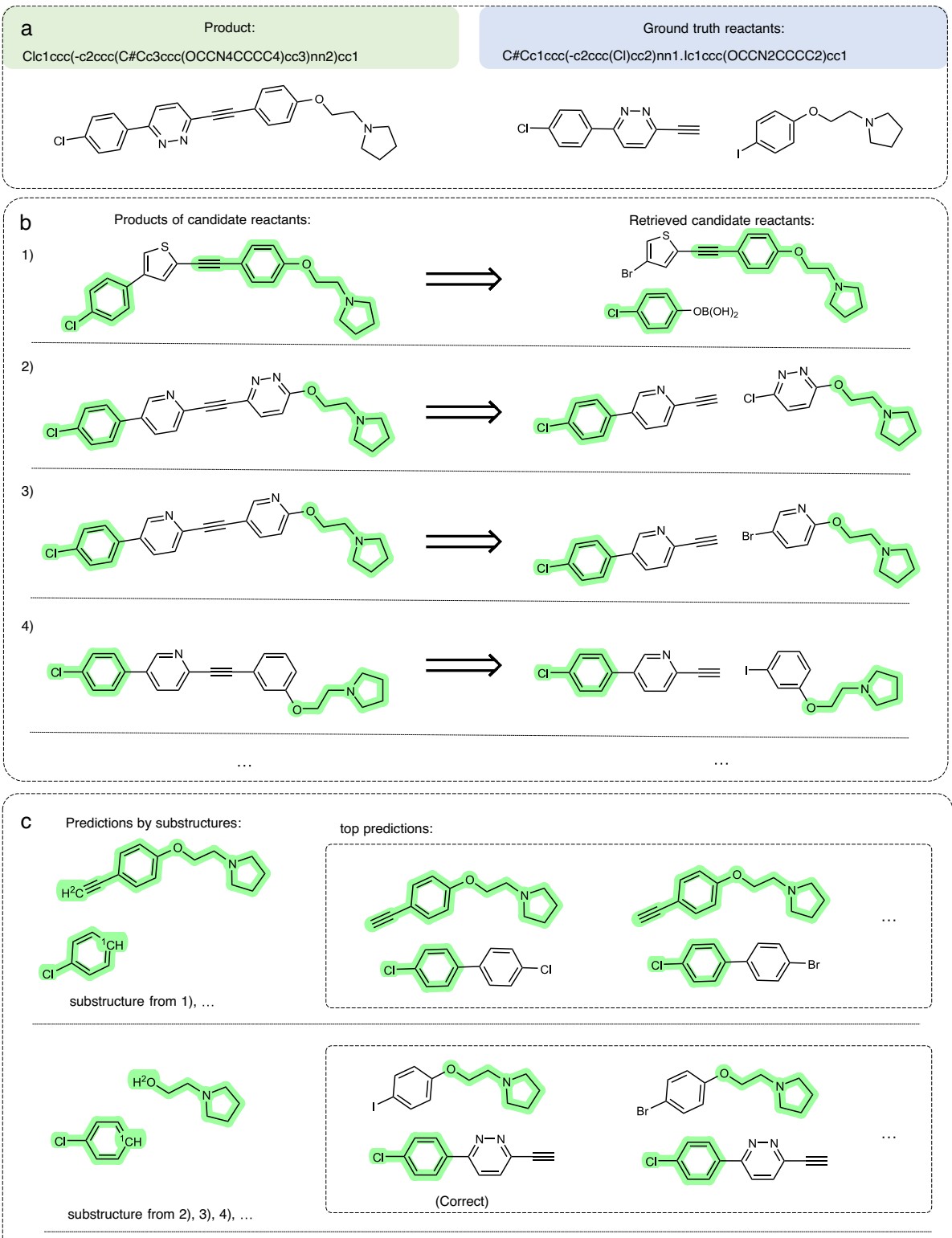

**Fig. 2 | Substructures and predictions grouped by substructures.** The retrieved candidate reactants (#2, #3 and #4) indicate that the substructures extracted from the retrieved reactant #1 is likely incorrect, because the triple bond is likely a reaction site. The extracted substructures are highlighted in green. **a** Product and reported reactants for the reaction. **b** Retrieved reactions. **c** Predictions from our model.

model has the tendency to break molecules down further than an expert chemist would, it can still fully reproduce the golden reactants when needed.

In summary, we developed a method to derive commonly preserved substructures to form the basis for retrosynthesis predictions.

The substructures are extracted using a fully data-driven approach with no human interventions. The overall approach is intrinsically related to how human researchers perform retrosynthesis analysis. Our current implementation achieved improvement over previously reported models. We also demonstrate that one way to improve

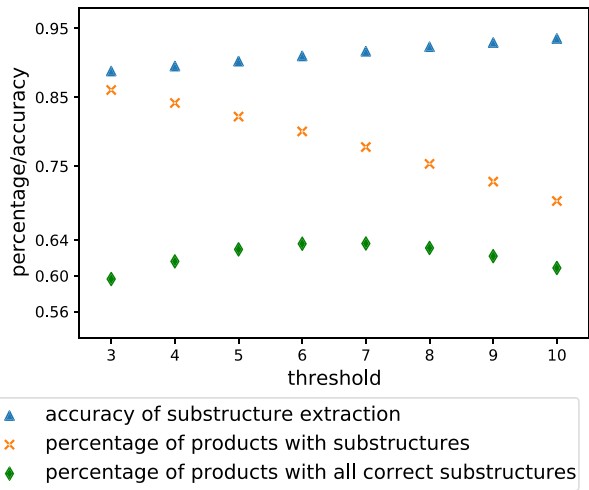

**Fig. 3 | Substructure extraction analyses.** The accuracy and the percentage of products with substructures under different thresholds.

retrosynthesis prediction for our model is by optimizing the underlying substructure extraction process. We hope this work will generate interest in this fast-growing and highly interdisciplinary area on retrosynthesis prediction and other related topics.

## Discussion

In this section, we address the advantages and disadvantages of our approach when compared with other existing approaches, which can broadly be classified as either template-based or template-free.

### Advantages over template-based approaches

Template-based approaches[18–22] use reaction templates or rule libraries, which contain information about atoms involved in reactions and chemical bonds near reaction sites. These methods heavily rely on the templates, which require considerable human effort to ensure that the template library covers most known organic reactions. They formulate retrosynthesis prediction as template classification or ranking problems[18–22] based on molecular similarity[15]. The resulting implementations involve deep neural networks to select top-ranked templates, which can then be applied to transform input molecules into outputs. The templates utilized in these methods usually depend on precomputed atomic mappings (how atoms in reactants map to those in the products). How to obtain a complete and reliable atomic mapping relationship is also a complex problem.

For test data with extracted substructures (about 82.2% of products from the whole test data), we reference the single-step retrosynthesis module in AiZynthFinder[22] as a representative template-based model for comparison. The single-step retrosynthesis module of AiZynthFinder was also trained on the USPTO data. The top 10 accuracy on template-based predictions is 62.9%, while our approach is 73.8%, which indicates that the coverage of the template library has room for improvement. We further test our model on the subset of data where golden reactants are not among the top 10 predictions of the template-based results, essentially singling out scenarios where template-based methods completely failed. The size of the subset is about 30% of the whole test data. For these scenarios, the accuracies of our model on the top 1, 5, and 10 predictions are 25.6%, 42.1%, and 46.7%, respectively. The accuracies are reduced because reactions in this subset are rare, not covered by reaction templates, or matched with incorrect templates. After checking template-based results on this subset, we discovered that for most predictions, the reaction sites predicted did not correctly match any templates in the library. Some incorrectly predicted reaction sites were included within the extracted

substructures by our model, which means that they actually remain unchanged during reactions.

Because substructures are usually inclusive of inactive parts of a molecule, we can use our model to narrow down possible options for locating potential reaction sites. Product molecule-specific substructures may help filter incorrect templates in template-based approaches; including well-known reactions in our approach can help extracting correct substructures. We leave integration of templates with our approach to have the best of both worlds as future work.

### Advantages over other template-free approaches

Template-free approaches can be categorized into graph edit-based and translation-based approaches. The graph edit-based approaches cast retrosynthesis prediction or reaction outcome prediction as graph transformations[16,23–25]. Modeling or predicting electron flow in reactions[26] can also be considered as a variant of graph-based methods. Besides, some semi-template-based methods also improve performance by identifying reaction sites followed by recovering graphs or sequences[5,27–29]. Translation-based approaches formalize the problems as SMILES-to-SMILES translation, typically with sequence models such as Recurrent Neural Networks[30] or the Transformer[2–4,31,32]. Variants of these approaches are introduced, such as reranking and pre-training[7,33]. Some models that fuse molecule graph information with translation-based approaches also achieved promising results[6,8]. Translation-based approaches can also be considered a two-step process: first locating possible reaction sites and then applying the "translation pattern(s)" learned from the data. The advantage of translation-based approaches over template-based approaches is that both steps are data-driven and translation patterns are inherently more flexible than templates.

Augmented Transformer[3] is a model that introduces random SMILES strings as data augmentation into their Transformer model, which we employ as a state-of-the-art baseline for comparison from the pool of template-free models. We built a subset of data for testing based on an overlap between products with successfully extracted substructures and golden reactants that were not among the top 10 predictions of the Augmented Transformer. The size of the subset is about 22% of the whole test data. For our model, the accuracies of top 1, 5, and 10 are 4.7%, 16.8%, and 22.9%, respectively. It is worth noting that this subset of cases is quite difficult to predict correctly. Compared with most template-free models that output SMILES strings representing complete reactant molecules, our model only generates SMILES strings of predicted fragments only. As shown in the section "Results on commonly preserved substructures", the average number of heavy atoms in the product, substructures, and golden reactants are 26.3, 12.1, and 30.0, respectively. Leveraging substructures have reduced difficulties by reducing the length of output sequence by approximately 40%.

We show a sample case in Fig. 5. The ground truth reaction is a Suzuki-Miyaura cross-coupling reaction that produces a four-ring hydrocarbon molecule. The product molecule contains no hetero atoms or functional groups other than aromatic hydrocarbons, which is not a common occurrence in the data set. The predicted reactants of the Augmented Transformer will not produce the given product in practice. While the structure of anthracene is covered by the substructure, we only predict the fragments of reactants (atoms or bonds that is not highlighted in Fig. 5(c)), which was easier compared to predicting the whole reactants' SMILES strings. The reactants predicted by our model indicate that the product can be obtained through combining benzene with anthracene via coupling reactions.

### Advantages of leveraging substructures

It is well-established that substructures and functional groups are essential concepts in chemical reactions. Reference[34] proposed graph

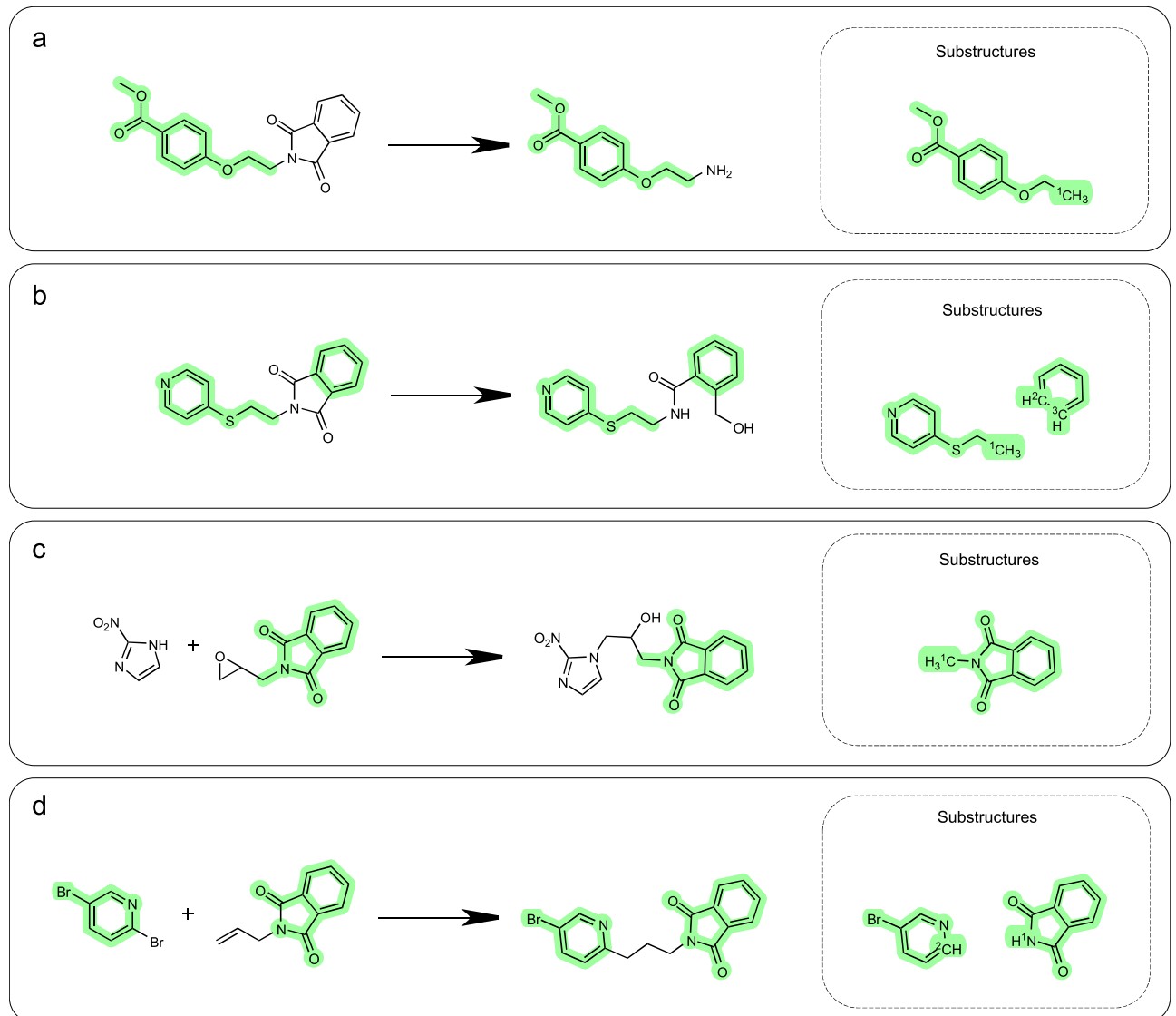

**Fig. 4 | Product molecule specific substructures. These reactants all contain phthalimide, with substructures highlighted in green. a** In this hydrolysis reaction, the methyl 4-ethoxybenzoate on the nitrogen atom is extracted as substructure. **b** In this reduction induced by sodium borohydride, the 4-(ethylsulfanyl) pyridine on the nitrogen atom and the benzene ring are extracted as substructure. **c** In this epoxy ring-opening process, the non-reacting phthalimide is extracted as substructure. **d** In this Heck coupling reaction followed by a reduction on the double bond, the heterocycle and the phthalimide are extracted as substructure.

motif-based self-supervised learning, where graph motifs refer to important subgraph patterns in molecules. The exploration of chemical substructures or subgraphs also provides efficient solutions to build large-scale chemical libraries[35] for drug discovery[36]. In our work, we explicitly introduce the concept of product molecule-specific structurally stable substructures for utilization in retrosynthesis predictions.

Commonly preserved substructures are expected to remain unchanged during reactions. A logical follow-up question is when considering reactants with multiple similar reactive groups, will non-reactive groups be correctly preserved in their substructures? We use amidation reactions as an example to perform a quantitative analysis. First, we aggregated products that were synthesized through selective amidation reactions on the test data with an additional requirement for reactants to contain more amine groups than the corresponding product molecule. We count the number of amide groups before and after the reactions. In this scenario, selective amidation reactions result in new amide groups being generated and the reactants

contained multiple active amine groups, introducing well-known chemistry concepts such as primary amines and secondary amines. In total, we analyzed 1,154 products. For our model, the overall accuracies of top 1, 5, and 10 are 60.5%, 80.2%, and 82.6%, respectively. The accuracy of extracted substructures is 90.6%. Among the set of correct substructures, 57.6% contained non-reactive amine groups. The result shows that a portion of the substructures are capable of preserving those non-reactive amine groups. Because substructures are extracted using a fully data-driven approach with no human intervention, it is possible that some atoms which remain unchanged during reactions are not included in the substructures. When we make predictions only with the substructures containing non-reactive amine groups, the accuracies of top 1, 5, and 10 are 67.7%, 85.6%, and 87.5%, respectively. This is another proof that if we were to select only chemically correct substructures by designing additional ranking or filtering models that incorporate existing chemistry knowledge instead of using all extracted substructures indiscriminately, prediction accuracies can be further improved.

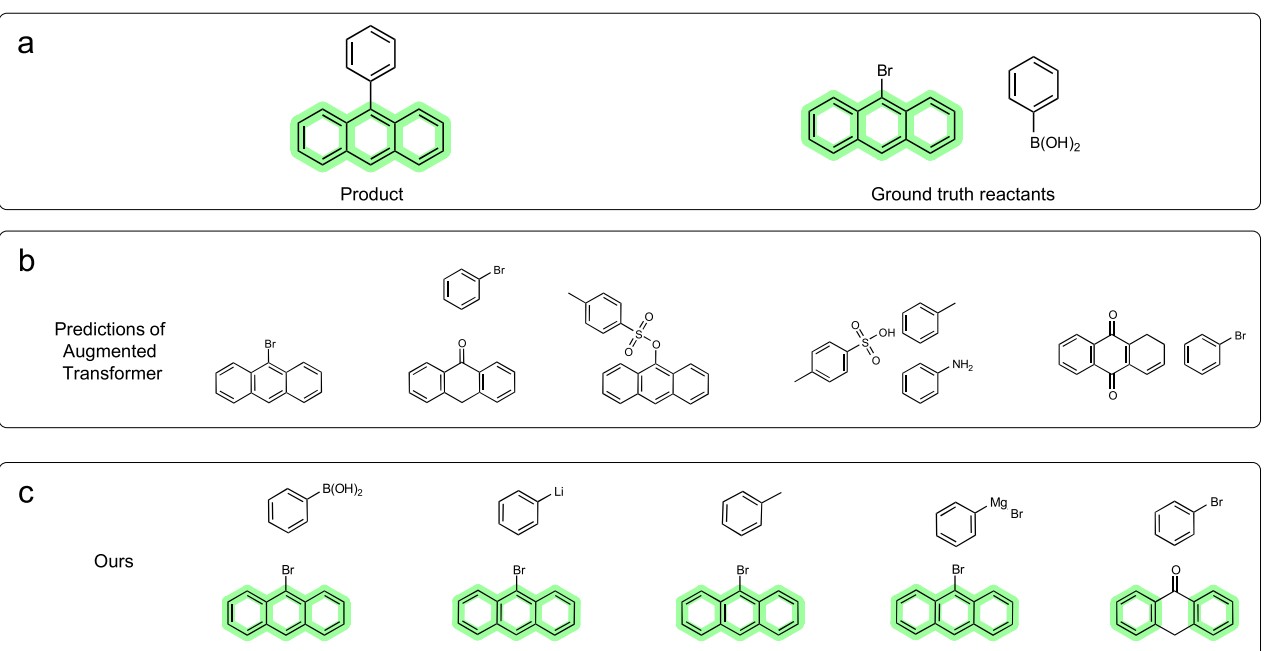

**Fig. 5 | Comparison with Augmented Transformer (substructures highlighted).**
**a** A polycylic aromatic hydrocarbons synthesized with Suzuki–Miyaura coupling reaction. **b** The predictions from Augmented Transformer[3], which cannot produce the expected product. **c** The predictions from our model, which gave ground truth reactants as the first prediction.

## Limitations

Our current model implementation successfully extracted substructures for about 80% of the training and test data set; the coverage clearly has room for improvement. The root cause for our model's failure to return any substructures for a given product molecule is that retrieved candidate reactants sometimes had no structural similarities. We speculate this to be attributed to either the dual encoder to not be fully trained, or the number of similar reactions for some product molecules remained limited. The results of Augmented Transformer, a very strong baseline we employed for single-step retrosynthesis prediction comparison, only achieved about 34% top 1 accuracy on products when no substructures are extracted. This indicates that the limited number of similar reactions for some product molecules is the main reason. We might need to collect more reaction data to improve the coverage in substructure extraction.

The substructures are extracted with a purely data-driven approach. Although we incorporated routines in our algorithms to preserve the most elementary chemical properties like aromaticity or stereoisomerism, there is no guarantee that the substructures we ultimately derived corresponded to known functional groups, nor can we use the substructures to explain the underlying reaction mechanism. As shown in the section "Results on commonly preserved substructures", some substructures obtained could provide hindsight for experts in selected cases.

Our method has an error propagation issue: incorrect substructures will increase probability of incorrect predictions. This can be mitigated by extracting substructures from all retrieved candidates, i.e., we can obtain correct predictions if the majority of extracted substructures are correct. The results observed so far indicate that performance can be further improved if accuracies of substructures are improved.

## Methods
### Reaction retrieval

In reaction pathway planning, chemists generally need to obtain insights and inspirations from existing reaction pathways learned through previous education and professional experience. Correspondingly, a retrieval module for a machine learning model must efficiently produce a list of candidates similar to the given query from a large collection of data. For retrosynthesis analysis, the query is the product, and the candidates are derived from "existing reactions", namely a large pool of reactions formed by the training and validation data.

To learn and measure the similarity between the reactants and the product, we used the dual-encoder architecture[14], which was introduced prior in memory-based machine translation[13]. We used two independent Transformer encoders[12] in the dual-encoder architecture, one to encode the reactants and the other to encode the products, as shown in Fig. 6.

Transformer[12] is a prominent encoder-decoder model that has achieved great success in natural language processing, computer vision, and speech processing. It consists of an encoder and a decoder, with each being a stack of $L$ identical blocks (highlighted using light yellow in Fig. 6). Each encoder block is mainly a combination of a self-attention module and a position-wise feed-forward network. Please refer to[12] for details about the Transformer model. Note that we only employed the Transformer encoder in our dual-encoder, without the decoder part.

We added the `[BOS]` token to the tokenized SMILES strings of both products and reactants, then fed them into the Transformer product and reactant encoders. The encoded output of `[BOS]` token represented the product and the reactants, denoted by $\mathbf{E}_{pro}$ and $\mathbf{E}_{rea}$, respectively. The overall objective is to minimize the distance between $\mathbf{E}_{pro}$ and $\mathbf{E}_{rea}$ in high-dimensional space for a given reaction. Following the training strategy proposed in[13], we developed two objectives for cross-alignment. The first objective was for golden reactants to have the highest-ranking score given the product, among all candidate reactants. This was approximated by maximizing the ranking score in a batch of product-reactants pairs when the batch size is relatively large. For a batch of $B$ product-reactants pairs sampled from the training set at each training step, let $\mathbf{X}$ and $\mathbf{Y}$ be the $B \times d$ matrix of the encoded product and reactants vectors,

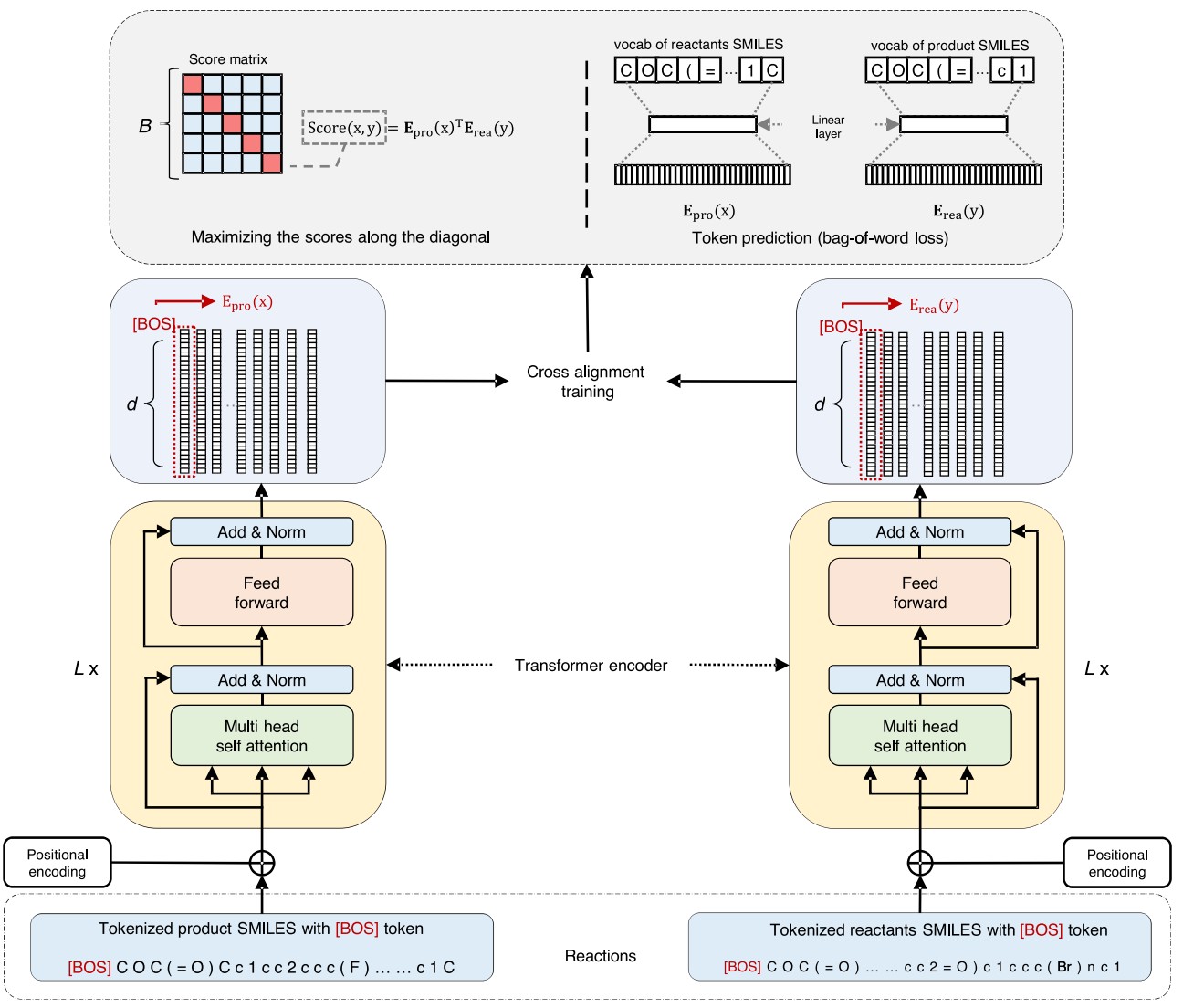

**Fig. 6 | The dual-encoder retrieval model. Both the product and reactant encoder have *L* identical blocks.** *x* is one product and *y* is one reactant(s) in the batch. The encoded output of the `[BOS]` token of *x* and *y* is denoted by a *d*-dimensional hidden vector $\mathbf{E}_{pro}(x)$ and $\mathbf{E}_{rea}(y)$, respectively. In one batch, the training objective is to maximize the scores along the diagonal of the score matrix and minimize the bag-of-words loss for token-level cross-alignment.

respectively, where $d$ was the hidden size. Each row in $\mathbf{X}/\mathbf{Y}$ corresponded to a product/reactant representation $\mathbf{E}_{pro}/\mathbf{E}_{rea}$. We defined $\mathbf{S}$, the ranking scores, as the dot product of the encoded product and reactants representations, namely $\mathbf{S} = \mathbf{XY}^T$, which was a $B \times B$ matrix of scores. Each row corresponded to a product and each column corresponded to a reactant in the batch. The pair $(\mathbf{X}_i, \mathbf{Y}_j)$ was considered aligned when $i = j$. The goal was to maximize the scores along the diagonal of the matrix and henceforth reduce the values in other entries. The loss function for the *i*th product-reactants pair was defined as:

$$\mathcal{L}_{rank}^{(i)} = \frac{-\exp(\mathbf{S}_{ii})}{\exp(\mathbf{S}_{ii}) + \sum_{j \neq i} \exp(\mathbf{S}_{ij})} \qquad (1)$$

The second objective was mainly borrowed from machine translation, which aimed to predict tokens in reactants' SMILES strings given the encoded product representations, and vice versa. This objective introduced additional semantic alignments between the product and reactant candidates at the token level. For the *i*-th product-reactants pair, the bag-of-words loss was used for this token-level cross-

alignment and formulated as

$$\mathcal{L}_{token}^{(i)} = -\sum_{w_y} \log p\left(w_y | \mathbf{X}_i\right) + \sum_{w_x} \log p\left(w_x | \mathbf{Y}_i\right) \qquad (2)$$

where $w$ represented one SMILES token of the product or reactants. The probability $p$ was computed by a linear projection layer followed by a softmax layer. For the dual-encoder model, the overall loss was

$$\mathcal{L} = \frac{1}{B} \sum_{i=1}^{B} \mathcal{L}_{rank}^{(i)} + \mathcal{L}_{token}^{(i)}. \qquad (3)$$

Once the dual-encoder was trained, we obtained dense vectors for all the reactants in the training and validation data. We leveraged Faiss[37], an open-source toolkit, to perform Maximum Inner Product Search (MIPS) on large collections of these dense vectors. The toolkit essentially built the index of dense vectors, which is optimized for MIPS search. The Faiss index code in our work was "IVF1024 HNSW32, SQ8", a graph-based index with a Hierarchical Navigable Small World (HNSW) algorithm[38]. In our approach, we pre-computed and indexed the dense vector representations of all reactants on the training and

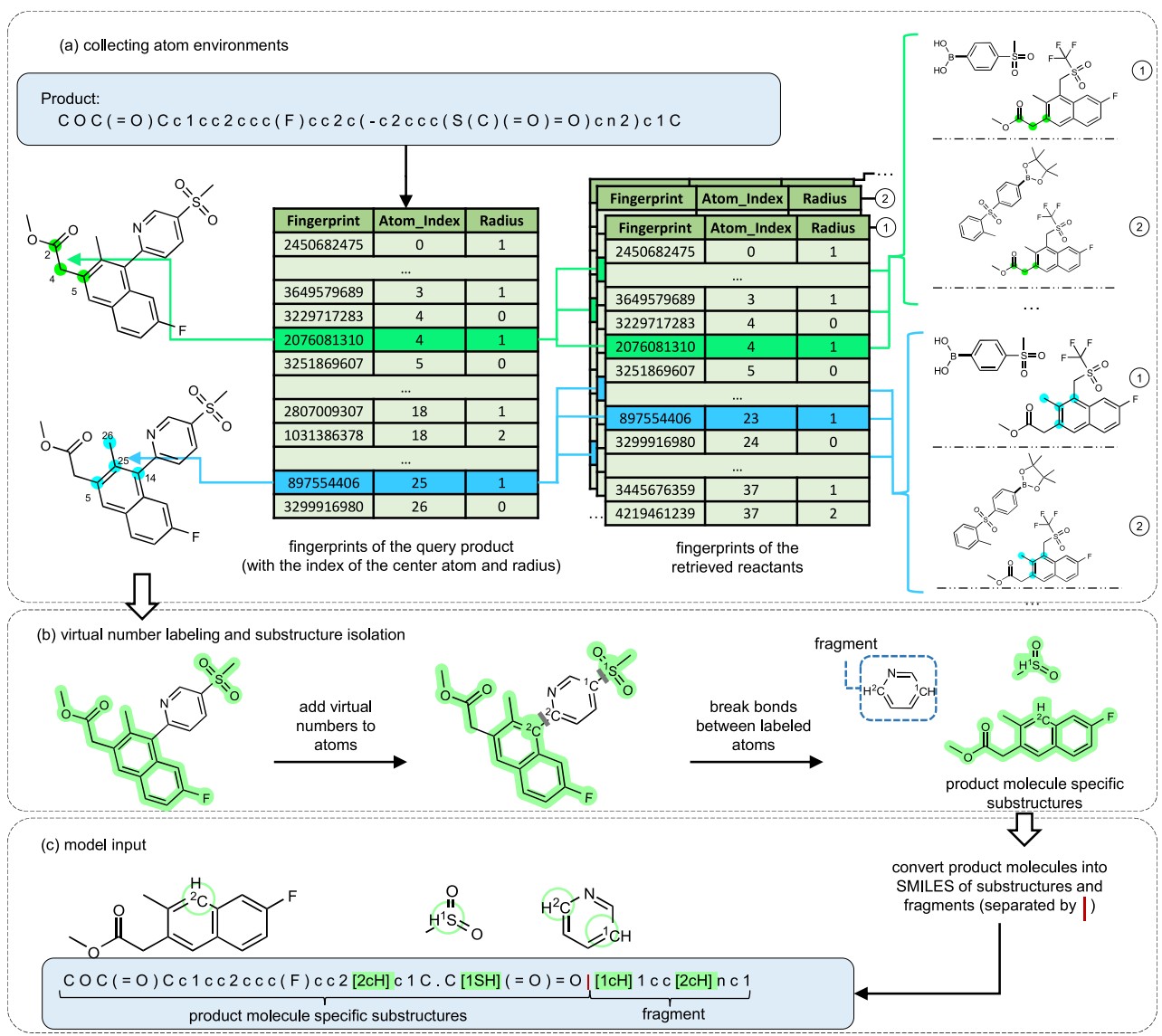

**Fig. 7 | Substructure extraction. a** Collecting atom environments, the atom environments of fingerprints `2076081310` and `897554406` in the product and the retrieved candidates are highlighted in green and blue, respectively. **b** Virtual number labeling and substructure isolation, the virtually labeled substructures are highlighted in green. **c** Model input, which is the SMILES strings of the substructures followed by the SMILES of fragments. The virtually labeled atoms are highlighted in green. "|" in the model input is a special character indicating the beginning of product fragments' SMILES strings.

validation data with the reactant encoder. For input product SMILES $x$, we used the product encoder to obtain its dense vector representation $\mathbf{E}_{pro}(x)$ and retrieved a ranked list of candidates by MIPS on the Faiss index.

**Substructure exaction**

Given the training objective of the dual-encoder model, the retrieved top candidates were expected to be similar to the golden reactants. We further assumed that these candidates shared a common substructure with the golden reactants. Although this hypothesis was not always valid, we observed that the assumption was reasonable in most cases. Common substructures were product molecule specific, because the retrieved candidates varied for each product molecule. Our goal was to extract common substructures given the product and its associated top cross-aligned reactants.

The extraction process was mainly based on molecular fingerprint, a widely used approach in molecular substructure extraction and similarity search. Using molecular fingerprints is one way to encode

the structure of a molecule. The most common type of fingerprint is a series of binary bits that represent the presence or absence of particular substructures in the molecule. Comparing fingerprints can help determine similarity between two molecules or locate aligned atoms. Using circular fingerprints is one of the methods capable of capturing 3D topological information. It maintains the environment of the center atom, which covers the neighbor atoms in different radii. The *de facto* standard circular fingerprints are the Extended-Connectivity Fingerprints (ECFPs), based on the Morgan algorithm, which is specifically designed for structure-activity modeling. Circular fingerprints are obtained through an enumeration of sub-molecular neighborhoods. First, each atom is encoded by an integer identifier, which is a hashed encoding representation of structural properties. The neighborhood information of the constituent atoms and bonds in different radii are iteratively assigned as the atom's numerical identifiers. The radius of a circular fingerprint refers to the size of the largest neighborhood surrounding each atom considered during enumeration. The fingerprint consists of the combination of all unique identifiers and is

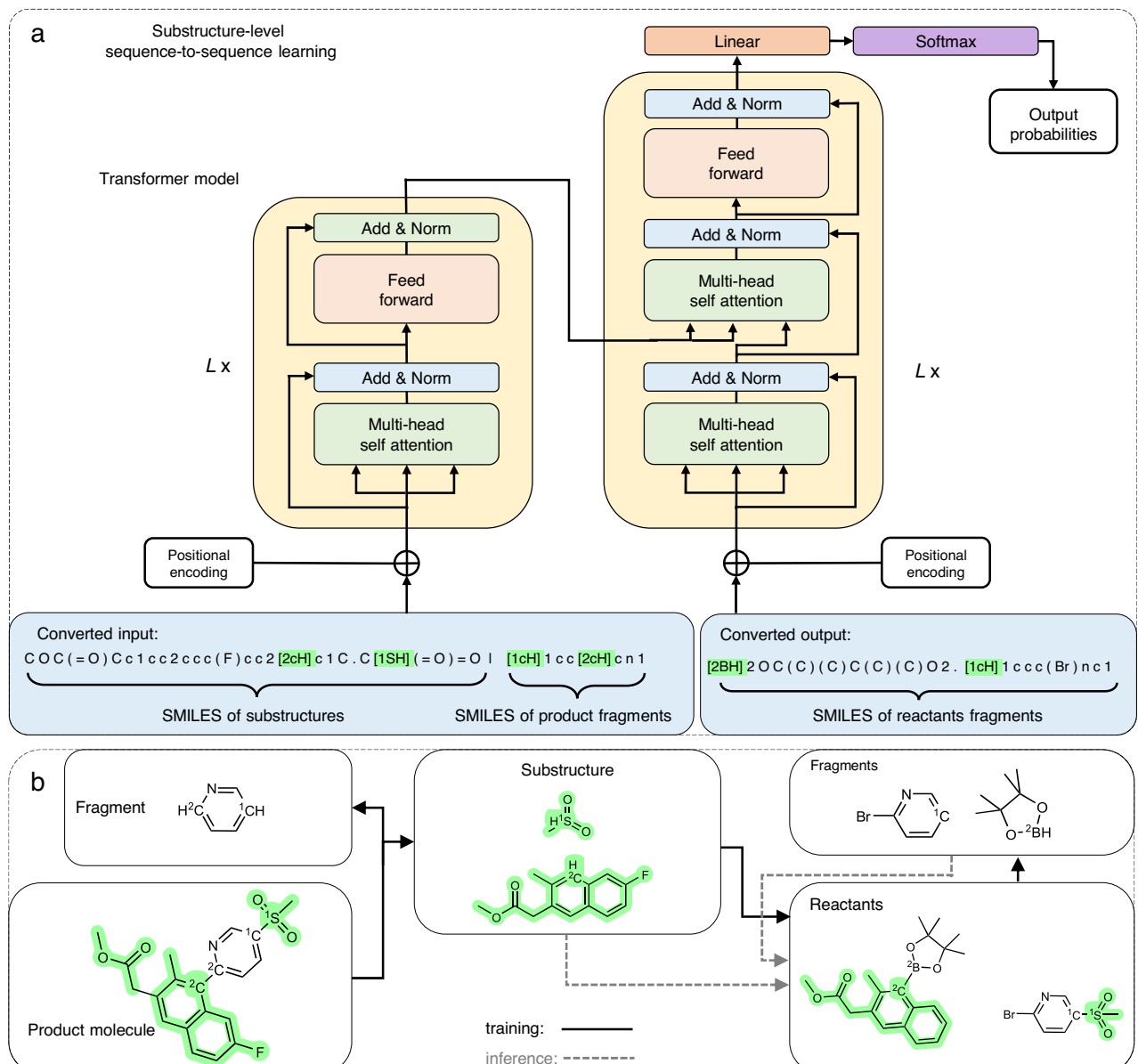

**Fig. 8 | Substructure-level sequence-to-sequence learning.** Both the Transformer encoder and decoder have $L$ identical blocks. The virtual number labeled atoms and substructures are highlighted in green. During training, the product side (input) is converted to substructures and fragments from the product, the reactants side (output) is converted to fragments from reactants only. "|" in the converted input is a special character marking the beginning of SMILES strings for fragments present in the product. The model is trained on the converted input and output. During inference, we only predict the fragments of reactants; finally predicted reactants are obtained by merging the substructures with the predicted fragments. **a** Substructure-level sequence-to-sequence learning model. **b** Our training and inference workflow.

subsequently folded into a binary vector of fixed length by converting integer identifiers into indices of the vector.

We used the toolkit RDKit[39] to extract common substructures. The overall extraction scheme is illustrated in Fig. 7. In our approach, we calculated the circular fingerprints of the product and the top 20 retrieved candidates with a radius ranging from 2 to 6. For example, in Fig. 7(a), the fingerprint 2076081310 encoded the environment of the center atom (index 4) and its neighbors with a radius of 1 in the product.

For each candidate reactants, we built the atom alignments with the product using shared fingerprints, as highlighted in green and blue in the fingerprint table in Fig. 7a. We selected atoms to build the substructure if they were aligned 5 times or more among the retrieved candidates. We further removed atoms in the substructure that were

aromatically bond to non-substructure atoms, or if connecting bonds showed stereoisomerism. Without these settings, splitting molecules into fragments was counterintuitive from a chemistry perspective, e.g., it may destroy aromaticity or stereoisomerism of the original molecule. For simplicity, we also removed atoms that were connected to multiple non-substructure atoms. Note that for a specific reaction, the atoms in the extracted substructure might not be fully connected. They could be different parts of one molecule or parts of different molecules, as shown in Fig. 7b. The extraction algorithm first used fingerprints to determine if an atom should be included in the substructure. Once the atoms of substructures were determined, all bonds between those atoms in the original molecule were kept as part of the substructure.

Next, we separated the product into substructures and other fragments. The assumption was that substructures tend to remain

unchanged during reactions. Note that it's possible for multiple fragments to be connected to atoms in the substructure. We introduced virtual numbers as labels to differentiate these bonds. As shown in Fig. 7(b), we added a virtual label to the bond between the atom S in the common substructure, and the atom c in the fragment, resulting in SMILES snippets with the virtual labels [1SH] and [1cH], respectively. Note that we purposely introduced additional hydrogen atoms in the substructure and other fragments after breaking the bonds, so they looked like charge-neutral molecules rather than radicals. We also recorded the type of bonds broken (double bond or triple bond) so that we can remove these hydrogen atoms easily when restoring the original molecule. Atoms with the same virtual number meant that they were connected in the original molecule before being broken down, for example, [1SH] was connected to [1cH] and [2cH] in the substructure was connected to the atom [2cH] in the fragment. With virtual number labeled bonds, we can easily isolate the substructure from other molecular fragments; we can also restore the molecule from the substructure and the associated fragments. The sites of broken bonds between the substructure and other fragments did not automatically translate to reaction sites in the chemistry sense; only a fraction of the sites were simultaneously reaction sites. For example, the broken bond between 1S and 1c was not a reaction site, as shown in Fig. 7.

It is worth noting that substructures were extracted using a purely data-driven approach with no human interventions other than the stereochemistry and aromaticity determination mentioned earlier on. Therefore, these substructures might not be "perfect" in the sense that they did not convey or correspond to specific chemical properties. "Perfect" is used to refer to cases when all broken bonds are reaction sites, and all atoms that remain unchanged are contained by the substructures. We will explain how we handle removed atoms or broken bonds that remain unchanged during reactions in the section "Substructure-level sequence-to-sequence learning".

To build a model that was not sensitive to the substructure for a given product molecule, we also extracted the center and neighboring atoms based on the common fingerprints as substructures from all the retrieved candidates. These substructures may be different as they come from different candidates. All the substructures from retrieved candidates that existed in the query (product) were used as input for model training and inference. Note that obtaining substructures based on common fingerprints between the product and the retrieved candidates might introduce errors because the retrieved candidates were not the golden reactants. We showed a sample of such error in the section "Results on commonly preserved substructures". For model training purposes, we can easily filter out incorrect substructures with golden reactants and only use correct substructures as the training data. The product molecule was represented multiple times with different substructures and fragments. This redundancy made output results robust over different substructures for a specific input product molecule. We were also able to group predictions by substructures to provide human experts with additional insights for decision-making in retrosynthesis planning compared with existing "black-box" models.

## Substructure-level sequence-to-sequence learning

Using methods detailed in the previous section, we isolated substructures on the reactant side of the training data. The product and reactants molecules were both converted into substructures and other molecular fragments. We used SMILES strings to represent these substructures and fragments and cast retrosynthesis analysis as substructure-level sequence-to-sequence learning problems. For sequence-to-sequence learning-based approaches, Molecular Transformer[2,3] achieved state-of-the-art performance on the reaction outcome prediction and retrosynthesis analysis[4] by employing textual SMILES representations of reactants and products. The model treated reaction prediction or retrosynthesis as a machine translation task. The

output SMILES was generated through a Transformer decoder token-by-token.

In our model, the input sequence was the SMILES string of substructures and fragments separated by "|", as shown in Fig. 1 and Fig. 8. "|" was a special character marking the beginning of SMILES strings for fragments present in product molecules. We assumed that substructures were stable and remained unchanged during reactions. For reactants, we only needed to predict virtually labeled fragments. Sometimes we failed to extract any substructures from a given product and its associated candidates retrieved due to the lack of atoms with the number of fingerprint alignments above the threshold. This happened because the minimum number of alignments required was set at 5 out of the 20 retrieved candidate reactants in our model. For these cases, the input was converted back to the original SMILES strings, with predictions performed by the data augmented Transformer model[3]. Based on our current implementation, the retrosynthesis analysis task was simplified and the average length of sequences to be predicted was significantly reduced compared to earlier models, which also helped reducing model complexities. Extracted substructures and predicted fragments containing virtual numbers enabled us to easily obtain the predicted reactants, as shown in Fig. 1. The structural changes among reactants and products were expected to be captured and predicted by the substructure-level sequence-to-sequence learning model. Note that for those atoms that remain unchanged during reactions but were not included in the substructures derived earlier, the model predicted them as output fragments.

Given an input product molecule, we extracted substructures from all associated candidates retrieved. The original product molecule was represented differently multiple times in this process, each time with a unique substructure and its corresponding fragments. During model inference, different substructures may lead to the same reactant molecules with different rankings. The ranking scores for the predictions of augmented SMILES strings were calculated mainly based on the output ranking by the beam search method[3]. In our implementation, the lengths of predicted fragments for different substructures were different, thus defining an empirical ranking formula based on the original beam search rank was not easy. As a result, we trained a pair-wise ranking model using a neural network with three linear layers on the validation data. The input features included frequency, ranking percentages among top 1 and top 2, average rankings on predictions of all the substructures and unique substructures. The training objective was set to ensure that on the validation data, the golden reactants had a higher score than incorrect predictions. See Supplementary Methods for more details on the pair-wise ranking model.

We tested our model on the USPTO_full benchmark with the same data split as[17], and performed evaluations using the top-$k$ accuracy of getting an exact match, i.e., given a product molecule, whether one of the top $k$ predicted reactants exactly matched the ground truth. Our model extracted substructures from about 80% of the training and test data. For a fair comparison, we trained a vanilla Transformer model with augmented random SMILES to obtain predictions for products with no substructures.

## Data and settings

We used the publicly available reaction data sets from the USPTO[40], which used SMILES strings to describe chemical reactions. We tested our approach on the USPTO_full benchmark with the same data split (train/valid/test) settings (80%/10%/10%) as[17]. There are approximately 1M reactions in total. We performed evaluations using the top-$k$ exact match accuracy, i.e., given a product molecule, whether one of the top $k$ predicted reactants exactly matched the ground truth. We canonicalized the molecules with the toolkit RDKit[39] and tokenized all the inputs following[30]. Note that the virtually labeled atoms were tokenized as single tokens. For each instance, we added two randomized

**Table 2 | Transformer parameter settings in the dual-encoder and the substructure-level sequence-to-sequence model**

| Parameters | Dual-encoder | Substructure-level seq-to-seq |
|---|---|---|
| Embedding size | 512 | 512 |
| Hidden size | 256 | 512 |
| Feedforward hidden size | 2048 | 2048 |
| Encoder blocks | 3 | 10 |
| Encoder attention heads | 8 | 8 |
| Max total training steps | 500,000 | 500,000 |
| Warm-up steps | 4000 | 8000 |
| Dropout | 0.1 | 0.1 |

SMILES strings as augmented data for substructures and fragments on the product side to be used in both model training and testing.

Table 2 shows the parameter settings. For the dual-encoder reaction retrieval model, following[13], we set the learning rate at 0.0001, the batch size at 4,096, and the label smoothing at 0.1. For parameters in substructure-level sequence-to-sequence learning, we mainly followed the Molecular Transformer settings[2]. We used the Adam optimizer[41] ($\beta_1 = 0.9, \beta_2 = 0.998$) and the Noam learning rate scheduler[12], where the scale factor was 2, and the number of warmup steps was 8,000. The batch size was 8,192 tokens, and the gradients were accumulated over 2 batches. For fair comparisons, we also trained a Transformer model with data augmentation (5 random + 1 canonicalized SMILES at the product side) under similar settings to obtain predictions for product molecules with no substructures extracted.

The dual-encoder was trained on the USPTO_full training data, the training process was stopped when the accuracy of alignment on the validation data was not improved. The ranking model had three linear layers of size 400; it was trained on the pairs collected from the validation data with label-smoothed cross-entropy loss. To collect training pairs, we trained our substructure-level sequence-to-sequence learning model on the USPTO_full training data, the training process was stopped when perplexity on the training data was no longer decreasing. Perplexity is a measurement of how well a model generates a sequence, which is a commonly used metric in machine translation and text generation. A lower perplexity indicates that the trained model is better at generating sequences, which suggests that it can be employed on the training data as an artificial metric to stop model training. Previous work[2,3] shows that the performance of Transformer-based models monotonically increases with the number of training steps, which suggests that we can use the perplexity on training data as the metric to stop model training. We used perplexity because when it has stopped decreasing on the training data, the model's capability at generating sequences cannot be further improved. We can obtain predictions on the validation data with the trained model. We further selected predictions that contained the golden reactants and paired the golden reactants with non-golden predictions. For each product, we collected at most 10 pairs of (golden reactants, non-golden reactants). There were about 510,000 pairs, and the training/valid split was set to 60%/40%. For each pair, we extracted ranking features for golden reactants and non-golden reactants to train the ranking model. The training objective was to ensure that golden reactants had a higher score than incorrect predictions. We stopped training when the accuracy on the validation data (40% of the pairs collected on USPTO_full valid data) was not improved. We used the scores generated by the ranking model to rank predictions on the USPTO_full test data.

After the ranker was trained, following Augmented Transformer[3], we combined the training and validation data of USPTO_full as the training set to train our model. We saved the model checkpoint every 10,000 steps, the training was stopped when perplexity was not decreased for 100,000 steps. Following Molecular Transformer[2], we

used average parameters of the last 10 checkpoints to obtain predictions on the test data. These predictions were further ranked by the scores generated by the ranking model.

## Data availability

The USPTO_full dataset is available at https://github.com/Hanjun-Dai/GLN. The processed data generated during and/or analyzed during the current study are available at https://github.com/fangleigit/RetroSub[42]. Source data are provided with this paper.

## Code availability

Codes and models are available at https://github.com/fangleigit/RetroSub[42].

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

## Author contributions

L.F. proposed to leverage commonly preserved substructures in machine learning models. L.F. and M.Z. designed and implemented the substructure extraction algorithm. J.L. performed deep analyses on substructures. L.T. provided constructive suggestions and feedback, including the idea of product molecule-specific substructures and manuscript writing. Research supervision: J.-G.L.

## Competing interests

The authors declare no competing interests.
