## [Peer Review File · Nature Communications]

Single-step Retrosynthesis Prediction by Leveraging Commonly Preserved SubstructuresREVIEWER COMMENTS

Reviewer #1 (Remarks to the Author):

The work subjected to my revision describes the new model for retrosynthesis, build as a concatenation of three (or four) models. In general, I find the idea of the work interesting and pushing state-of-the-art results further, nevertheless, I think that more understanding of model results is needed.

In particular, one of the model parts is the "Reaction-aware Substructure Exaction". First of all, I don't see where there "reaction-awareness" comes in. By "reaction-awareness" I understand, that the model somehow understands, what reaction should be used to obtain a given product. But this is not the case. The model finds the template reactions based on the similarity of fingerprints. As far as I understand, there is no ranking of possible reaction types, which could lead to desired product. Therefore, I find the logic, and the title itself highly confusing.

Also, although I regard the technique interesting, the aim of the this submodel is mainly to extract the templates. This rises two questions. First is, is the model recalculating the fingerprints when searching for the templates? Because if it does so, it would highly decrease its speed, compared to a set of precomputed fingerprints. Second, did the authors checked, what is the model performance on the cases, where the standard template synthesis fail? Actually, the retrosynthesis made by chemists follows the plan: 1) find similar reactions; 2) extract the reaction center and compare to desired product; 3) understand, what can fail in the predicted reaction given, that the predicted substrates are different than in the reference reactions. The first two steps are covered by the algorithm presented. But is the third step? In particular, how does the model behave on multistep (one-pot) reactions, or on reactions, where there are more than one similar reactive groups? If the target reaction is amidation, does the model propose to block the non-reactive amine group? Because of the construction of the model, I doubt it, but I may be wrong and this should be discussed by the authors.

In general, the strenghts and the weak points of the model should be discussed. In particular, obtaining the similar structures by comparing the circular fingerprints is obviously viable to problems, when comparing structures, which are different structuraly, but similar chemically, like e.g. amides and lactames. Or vice versa - upon substitution of heterocycles, their properties may change drastically, although the relative change in structure is low. This should be discussed and analyzed.

Also, the chemistry obtained by the authors is doubtfull. In the Fig. 7c there is reaction which is labelled the Functional group interconversion. Actually it is not. The mechanism of this reaction is the attack of the lone pair of the imidazole ring on the epoxyde ring with opening of the latter. As a result, the reaction aware substructure should span until the carbon atom connected to the hydroxyl group, if the correct reaction mechanism should be taken into consideration. And it should, if there should be any "reaction-awareness". Also, when discussing this Figure, there is no reduction in panel B - no atom changes its oxidation state. Moreover, in this reaction in the reaction specific substrates there is an order benzene ring with two labelled atoms, but actually the hydroxyl group also do not take part in this reaction, and therefore there should be the whole phenole (hydroxybenzene).

When talking about the labelled atoms - the authors name them "isotopically labelled atoms". I find

this name very confusing. "Isotope" is a well established name in chemistry, indeed used in NMR to label atoms. But in this case, the authors do not label atoms, but rather bonds. In particular, they label two different atoms with the same number, even if the atoms represent different elements. This should not be named "isotopes".

In general, the naming is a bit confusing. The authors use the notion of "target" to name the substrates, while the product of the reaction, commonly called target of the reaction is called "source". Also, the authors use the notion of "reactants" instead of substrates. This suggest, that the authors take into account the solvent and the catalyst apart of the substrates. Is that the case? Please specify. And if so, the USPTO dataset is a collection of patents, which have to be preprocessed to be used in the training. How are they preprocessed? How do the authors obtain the smiles?

Apart from this, there are a few places, where additional information is needed. In particular, the authors add, that to order the reaction correctly, they "train a pair-wise ranking model". The model is described very shortly, but actually it is highly important, as it orders the reactions and have a tremendous influence on the Top1 and in general TopK metric. In particular, to separate out the model and the ordering, the authors should calculate the TopK for their model without and with the ordering. Only this would tell, if the superior values in Tab. 2 are due to model construction, or due to the trained ordering.

Finally, the figures are not helping much with the current captions. Take for example Fig. 4. There are some panels there, arrows, the block scheme of the whole transformer model, some substructures are highlighted with green (which actually hinders resolving the atoms), but these do not match the substructures in smiles, there is some misterious Lx, and all of these is described with only "Substructure-level sequence-to-sequence learning". I spent a lot of time trying to understand, what the Figures are really presenting and how to match it with the manuscript.

To sum up, although I find the idea of the work interesting, I think that much more work is needed to turn it into full paper - one has to understand the drawbacks of the model and discuss them, present the idea in a more clear way, separate out what is the effect of the model and what of reranking, understand the chemistry it produces, and finally compare its identified features and drawbacks with other models. By the way, the other models also produce top N predictions, and therefore are not as "black-box" as the authors claim.

Reviewer #2 (Remarks to the Author):

The problem presented in the work is well stated and is surely of scientific interest to both the Chemistry community and the AI community. I like the approach of starting from a chemists-intuition on how to perform retrosynthesis and then putting it in practice through an innovative approach. I also appreciate the template-free nature of the approach. The level of detail is appropriate.

Syntax and general paper structure:

1. I would delete the last sentence of the abstract, the one starting with "We hope ...". It is more for the "Conclusions".

2. Introduction: line 20 -22, there is a reference missing (Schwaller et al. 2020, <https://pubs.rsc.org/en/content/articlelanding/2020/sc/c9sc05704h>)

3. Maybe a short summary of the sections in the paper and what they contain should be added at the end of the "Introduction".

4. Line 128 specify better equation (2) and what is `w`.

5. The data section, line 246, should be moved somewhere else, like in 'Methods'.

6. Figure 6, specify "products with CORRECT substructures" and "Products with ALL CORRECT substructures" in the legend.

7. The section from line 341 on should be the first chapter in "Results".

8. There is no "Conclusion" section which makes it hard for the reader to understand the main findings and impact for the work (you need to read between the lines).

Work results:

9. How is the validation set used? Hyperparameters search or just checkpoint selection? Should be more specific for both the "Reaction retrieval model" and the "Substructure-level sequence-to-sequence learning" on how the validation is performed.

10. In paragraph starting from line 141: Suppose you count more than 5 occurrences for a chain of 4 carbon atoms and then you also count more than 5 occurrences for the SAME CHAIN with one atom more (ex. oxygen). You keep both substructures or just the smaller/bigger one? It would be nice to be more specific on this.

11. Line 143-145. It should be shown how you observe that this assumption is reasonable, with statistics.

12. Line 186-188. Is there a reason for the choice of representation with charge-neutral molecules instead of radicals?

13. Line 188-190: Where and how is the single/double/triple bond information stored if the bond is cleaved?

14. Line 198-207: I disagree on this. I like this augmentation approach, but for the training set only the golden reactants should be used, for example using different radii for the substructure. If all the retrieved reactants are used, considering that some bring incorrect substructures you just introduce more noise in the training. The reaction retrieval model should be used only at test time.

15. Line 238-243: Not sure I understand why you need a ranking model. Can you just use some "majority" ranking where the top-1 prediction is the one more frequent among the predictions from different substructures from the same target? (After reconstructing the reactants of course).

16. Line 252-254: Is this on top of "substructure augmentation"? So, for each product, for each of its substructure's representations You add 2 randomized SMILES (ex. If I have 5 different valid substructures for a product -> 15 representations for the same product in the training set?).

17. Line 277: see previous comment: for training substructures should be extracted only from the golden reactants.

18. From line 341 on: are in Table 2 the top-x accuracies from the "substructure-found" test set combined with the "no-substructure-found" test set? They should be split as well.

19. What is the percentage on invalid SMILES prediction compared to other models?

We appreciate the careful, detailed, and chemistry-oriented feedback provided by the reviewers, who have substantially more chemistry field knowledge than we do. We also realized that the terms we used in our original manuscript could carry confusing meanings to a chemistry audience, so we changed the terminologies used throughout the paper, and tailored our revised manuscript accordingly so that it more precisely delineates what we are trying to do, making it potentially interesting for a chemistry audience. We would like to share our early results in this fast-growing and highly interdisciplinary area, we do realize the current shortcomings of our approach, which we intend to improve further in our subsequent work. We thank the reviewer's sincere effort in giving feedback.

Reviewer #1 (Remarks to the Author):

The work subjected to my revision describes the new model for retrosynthesis, build as a concatenation of three (or four) models. In general, I find the idea of the work interesting and pushing state-of-the-art results further, nevertheless, I think that more understanding of model results is needed.

In particular, one of the model parts is the "Reaction-aware Substructure Exaction". First of all, I don't see where there "reaction-awareness" comes in. By "reaction-awareness" I understand, that the model somehow understands, what reaction should be used to obtain a given product. But this is not the case. The model finds the template reactions based on the similarity of fingerprints. As far as I understand, there is no ranking of possible reaction types, which could lead to desired product. Therefore, I find the logic, and the title itself highly confusing.

Response 1: the term "reaction-aware" in our original manuscript was used because the substructures were extracted from the reactions that were structurally similar to the potential reaction to obtain the given product molecule. Our approach output different substructures for different products without a prior template library (which we interpret to incorporate existing chemistry knowledge by experts). After a careful reconsideration of the reviewers' input, we realized that using the term "reaction-aware" is unsuitable because it overpromised what the model is capable of achieving in its current state, and the substructures extracted by our approach is solely based on the USPTO dataset that only uses SMILES string to describe chemical reactions. We do not incorporate chemistry knowledge that is familiar to most chemistry-oriented readers or is inherently present in some of the template-based approaches. Our algorithm is unable to derive sub-molecular, or sub-atomic first-principle chemistry mechanisms due to this limitation. We are trying to mostly demonstrate that without any inherent field knowledge to understand the reactions present in the USPTO dataset, a data-driven model is still capable of producing useful trends and predictions based on reactions described by SMILES strings. The results show that the substructures extracted by our approach indeed showed the "less reactive", or more precisely, "commonly preserved substructures" or "product molecule specific substructures" that are potentially interesting to chemists.

Also, although I regard the technique interesting, the aim of the this submodel is mainly to extract the templates. This rises two questions. First is, is the model recalculating the fingerprints when searching for the templates? Because if it does so, it would highly decrease its speed, compared to a set of precomputed fingerprints.

Response 2: our model does not have a reiteration (recalculating) process after the substructures are extracted. We do not have a method to re-evaluate the quality of the extraction process itself based on the feasibility to react or expert field knowledge.

Because the fingerprints of all reactions on the train and dev data could be precomputed in one go and then aggregated to form the reaction library, we only need to calculate the fingerprints of the input product molecule, the overall time should be comparable to template-based approaches.

Second, did the authors checked, what is the model performance on the cases, where the standard template synthesis fail?

Response 3: we performed additional analysis based on the reviewer's feedback, with the open-source template-based retrosynthesis tool AiZynthFinder¹. We only used the single-step retrosynthesis module in AiZynthFinder, which is also trained on the USPTO data. We tested our approach on the test data, where the golden reactants are not in the top 10 predictions of AiZynthFinder, essentially singling out scenarios where template-based methods completely failed. For these scenarios, the accuracies of our approach on the top 1, 5 and 10 predictions are 25.6%, 42.0%, and 46.6%, respectively. We added detailed discussions of advantages over template-based approaches in Section 4.1 of the revised version.

Actually, the retrosynthesis made by chemists follows the plan: 1) find similar reactions; 2) extract the reaction center and compare to desired product; 3) understand, what can fail in the predicted reaction given, that the predicted substrates are different than in the reference reactions. The first two steps are covered by the algorithm presented. But is the third step?

Response 4: the third step is not covered by our algorithm. Understanding failures requires deep chemistry knowledge to filter out those predicted reactions which have higher probabilities to fail, before attempting an experiment to confirm the real-world result. In our approach, we grouped the predictions by substructures extracted from the reactions (an example is shown in Figure 5), which provides user with substantial information to assess which predicted reaction might fail. We believe our paper's contribution to the third step is that when compared to existing approaches that provide predictions only, the substructures and the retrieved reactions make the third step less difficult to an expert.

Note that on the USPTO_full benchmark, template-free approaches achieved state-of-the-art performances, however, most of them do not follow how humans perform retrosynthesis (none of the above steps are incorporated). Therefore, the methods are not intuitive from a chemistry perspective. Our work could be considered as one step towards adding a chemist-like intuition to machine learning. We value your suggestions plan to assess the third step in future work.

In particular, how does the model behave on multistep (one-pot) reactions, or on reactions, where there are more than one similar reactive groups? If the target reaction is amidation, does the model propose to

¹ <https://github.com/MolecularAI/aizynthfinder>

block the non-reactive amine group? Because of the construction of the model, I doubt it, but I may be wrong and this should be discussed by the authors.

Response 5: we did not tackle multistep or one-pot reactions in this work, our evaluation is that performing these will be exceedingly difficult on the USPTO_full data, which were based solely on single-step reactions.

For reactants with multiple similar reactive groups, we performed additional analysis, using amidation reactions on the USPTO_full test data as an example. In our results, we found that 57.6% of the time, the non-reactive amine group will be blocked (were correctly preserved in our substructures). We added the detailed analysis in Section 4.3 of the revised version.

In general, the strengths and the weak points of the model should be discussed.

Response 6: we added Section 4 to discuss the strengths and weak points of our model in the revised version.

In particular, obtaining the similar structures by comparing the circular fingerprints is obviously viable to problems, when comparing structures, which are different structurally, but similar chemically, like e.g. amides and lactams. Or vice versa - upon substitution of heterocycles, their properties may change drastically, although the relative change in structure is low. This should be discussed and analyzed.

Response 7:

We understand the reviewer's question to mean that for a chemist who is naturally aware of the underlying chemical mechanisms, sometimes a similar looking structure means the chemistry mechanism are drastically different, or vice versa (significantly different looking reactions from a structural point of view actually involve similar reaction mechanism), is our machine learning model able to delineate these basic chemistry concepts.

We have revised our manuscript to say that our extraction algorithm captures those substructures that tend to be preserved in a chemical reaction. Our approach relies on both preserved substructures and the fragments that were subtracted from the molecule. For a specific product molecule, the input was the SMILES strings containing both the substructure and the fragments, which were jointly encoded with a Transformer network. Transformer is a powerful sequence-to-sequence learning model that has been extensively used in natural language processing (NLP), and the core emphasis in the NLP application is about context, that means, the word, when placed in different contexts, can have drastically different meanings. For our task, the output token is decoded based on the contextual representations of both the input substructure and fragments. Analogous to NLP problems, the subtle structural or chemical similarity/difference is expected to be captured by the Transformer model.

Our initial work in this paper did not include comparison of structures for an in-depth analysis related to underlying mechanisms, because performing such an analysis will require additional data that is beyond the scope of information available in the USPTO dataset.

Our current work uses a fully data-driven approach. Although we incorporated routines in our algorithms to preserve the most elementary chemical properties, like aromaticity or stereoisomerism, there is no guarantee that the substructures we ultimately derived correspond to known functional groups, nor can we use the substructures to explain the underlying reaction mechanism.

Also, the chemistry obtained by the authors is doubtful. In the Fig. 7c there is reaction which is labelled the Functional group interconversion. Actually it is not. The mechanism of this reaction is the attack of the lone pair of the imidazole ring on the epoxyde ring with opening of the latter. As a result, the reaction aware substructure should span until the carbon atom connected to the hydroxyl group, if the correct reaction mechanism should be taken into consideration. And it should, if there should be any "reaction-awareness".

Response 8: the reaction types in our original manuscript were derived following Schneider et al. [Ref_1]. In that paper, the authors classified the reactions into 10 categories: heteroatom alkylation and arylation, heterocycle formation, acylation and related processes, deprotections, functional group addition, reductions, protections, C-C bond formation, functional group interconversion, oxidations; for a well-trained chemist, the categorization is rather vague. With the reviewer's input, we removed the vague categorization schema from our manuscript.

As we explained in Response 7, the substructures we extracted using a fully data-driven approach with no human intervention, and is not chemically aware of functional groups known to human experts, or even an epoxide ring explicitly taught in basic organic chemistry. More explicitly, the algorithm is not aware of the uniqueness, and therefore, the specific considerations that should be given to those two epoxide carbons, and how an epoxide carbon should be given more respect compared to a normal carbon during bond formation/breakage. Because our algorithm does not incorporate the underlying knowledge or appreciate the uniqueness of human discoveries, but merely assesses bond breakage in a brutal and elementary manner to obtain substructures that remain unchanged during reactions. In our current extraction algorithm, one atom will be included in the substructures if it is among the environment of a common fingerprint of minimal radius 2. That's why the carbon atom connected to the hydroxyl group is not included in the substructure. In our planned later work we should probably make sure that well-established functional groups are given proper respect for their uniqueness during substructure extraction.

[Ref_1] Nadine Schneider, Daniel M Lowe, Roger A Sayle, and Gregory A Landrum. Development of a novel fingerprint for chemical reactions and its application to large-scale reaction classification and similarity. *Journal of chemical information and modeling*, 55(1):39–53, 2015

Also, when discussing this Figure, there is no reduction in panel B - no atom changes its oxidation state. Moreover, in this reaction in the reaction specific substrates there is an order benzene ring with two labelled atoms, but actually the hydroxyl group also do not take part in this reaction, and therefore there should be the whole phenole (hydroxybenzene).

Response 9: The reaction is classified as reductions mainly due to the vague categorization schema which we explained in Response 8. With the reviewer's input, we removed the vague categorization schema from our manuscript.

Similarly, if the algorithm is unable to "discover" the unique reactive properties of an epoxide ring (See Response 8), it definitely cannot discover the uniqueness of a phenole, or that a benzene ring by itself, is one of the most chemically stable structures in nature, and extremely unlikely to be a reactant, unless it has other functional groups already attached to it. In our planned future work, we should probably make sure that well-established functional groups are given proper respect for their uniqueness during substructure extraction.

When talking about the labelled atoms - the authors name them "isotopically labelled atoms". I find this name very confusing. "Isotope" is a well established name in chemistry, indeed used in NMR to label atoms. But in this case, the authors do not label atoms, but rather bonds. In particular, they label two different atoms with the same number, even if the atoms represent different elements. This should not be named "isotopes".

Response 10: Thanks for pointing this out. In our paper, our original terminology of "isotopic number labels" are indeed confusing to a chemistry audience because they do not actually denote changes/differences in the number of neutrons in an atom's nuclei, where we only meant to use the numbers for tracking sites of bond breakage between substructures and fragments. In order not to confuse with the isotope's chemistry definition, we changed "isotope number" to "virtual number", "isotopically labeled atoms" to "virtually labeled atoms".

In general, the naming is a bit confusing. The authors use the notion of "target" to name the substrates, while the product of the reaction, commonly called target of the reaction is called "source". Also, the authors use the notion of "reactants" instead of substrates. This suggest, that the authors take into account the solvent and the catalyst apart of the substrates. Is that the case? Please specify.

Response 11: Thanks for pointing this out, too. We really appreciate this. In machine translation or text generation, "source" was defined as the model input, while "target" is defined as the output, e.g., translate from the source language to the target language. Indeed to a chemist this is confusing because as the reviewer mentions, the product of a series of chemical reactions is conventionally known as the target. In order not to confuse machine learning terminologies for NLP translation to chemistry terminologies, we changed the "target" in our paper to "reactants" and "source" to "product". This hopefully makes it clear that our model input is the product, while the model output/predictions are the reactants.

We also realized that the reviewer was quite correct in pointing out that solvents and catalysts play a crucial role in how a reaction proceeds from reacting species to product species; the choice of solvents/catalysts not only changes what's ultimately formed as the product, but also reaction rates/mechanisms. In the current version of our model, we did not take solvent and catalyst into account. This was done because we observed that previous retrosynthesis papers such as AugmentedTransformer[Ref_2] defined "reactants" and "reagents" differently from their strict chemistry definitions. This obviously does not necessarily reflect how chemistry works in the real world. We agree that these simplifications are problematic in predicting certain reactions where

solvents/catalysts play crucial roles in determining what ultimately forms as the product. We decide to follow the other authors definitions, where the reactants are defined as chemical species that contribute atoms to the product (we believe organic chemists refer to these as substrates). The rest of the chemical species involved in a reaction (e.g., solvents and catalysts), but do not contribute any atoms to the product, are not considered in our model. We tried to make this deviation from standard chemistry terminology clear in the revised manuscript.

[Ref_2] Tetko, I. V., Karpov, P., Van Deursen, R. & Godin, G. State-of-the-art augmented nlp transformer models for direct and single-step retrosynthesis. *Nature communications* 11, 1–11 (2020).

And if so, the USPTO dataset is a collection of patents, which have to be preprocessed to be used in the training. How are they preprocessed? How do the authors obtain the smiles?

Response11: Indeed we did not use the USPTO data as is, but relied on the extracted/preprocessed data from previous contributors in this field. Initially, the patents are processed by the authors in [Ref_3], where the patents were extracted for information about reactants, products, and an incomplete set of related info (yield, temperature, pressure, solvent, catalyst) that were sporadically included. The USPTO_full benchmark used in this paper was released by Dai et al. [Ref_4]. For the reactions extracted from patents [Ref_3] with multiple products, Dai et al. [Ref_4] separated these reactions so that each product was a separate entry containing the same reactants. After removing the duplicates and reactions with wrong atom mappings, roughly 1M unique reactions remained, which were further divided into train/valid/test sets containing 800k/100k/100k reactions, respectively.

[Ref_3] Lowe, Daniel (2017): Chemical reactions from US patents (1976-Sep2016). figshare. Dataset. <https://doi.org/10.6084/m9.figshare.5104873.v1>

[Ref_4] Hanjun Dai, Chengtao Li, Connor Coley, Bo Dai, and Le Song. Retrosynthesis prediction with conditional graph logic network. *Advances in Neural Information Processing Systems*, 32, 2019.

Apart from this, there are a few places, where additional information is needed. In particular, the authors add, that to order the reaction correctly, they "train a pair-wise ranking model". The model is described very shortly, but actually it is highly important, as it orders the reactions and have a tremendous influence on the Top1 and in general TopK metric. In particular, to separate out the model and the ordering, the authors should calculate the TopK for their model without and with the ordering. Only this would tell, if the superior values in Tab. 2 are due to model construction, or due to the trained ordering.

Response12. The pair-wise ranker only slightly improved the overall performance. We added some details in the Supplementary Methods and Supplementary Results sections of the revised manuscript. The improvements in our model compared to previous work were mainly attributed to our implementation of substructures.

Finally, the figures are not helping much with the current captions. Take for example Fig. 4. There are some

panels there, arrows, the block scheme of the whole transformer model, some substructures are highlighted with green (which actually hinders resolving the atoms), but these do not match the substructures in smiles, there is some mysterious Lx, and all of these is described with only "Substructure-level sequence-to-sequence learning". I spent a lot of time trying to understand, what the Figures are really presenting and how to match it with the manuscript.

Response 13: we add brief descriptions to all the figures in our paper. Throughout the paper, we highlighted all virtually labeled atoms and substructures in green. For Fig.4, the converted input is the SMILES of the substructure and fragments, separated by "|", where "|" is a special character indicating the start of product fragments SMILES. Thus, the SMILES of the substructures is the content before "|" in the converted input. "Lx" means there are L identical transformer blocks in the encoder or decoder. We made all figures more clear in the revised version.

To sum up, although I find the idea of the work interesting, I think that much more work is needed to turn it into full paper - one has to understand the drawbacks of the model and discuss them, present the idea in a more clear way, separate out what is the effect of the model and what of reranking, understand the chemistry it produces, and finally compare its identified features and drawbacks with other models. By the way, the other models also produce top N predictions, and therefore are not as "black-box" as the authors claim.

Response 14: We addressed most concerns in the previous responses. For deep neural networks, "black box" means we don't know how all the individual neurons work together to arrive at the final output, and it isn't even clear what any particular neuron is doing on its own. Transformers could be considered as "black box" models. Our approach tries to add some chemist intuition to machine learning models, which makes the model more explainable compared to existing models. In addition, our model extracts substructures that tend to suggest low reactivity, and thus portions of a reactant/substrate molecule that tends to get preserved in the product molecule. The model does this entirely without expert interventions but yields information that could be useful to assist an expert in reaction planning.

Reviewer #3 (Remarks to the Author):

The problem presented in the work is well stated and is surely of scientific interest to both the Chemistry community and the AI community. I like the approach of starting from a chemists-intuition on how to perform retrosynthesis and then putting it in practice through an innovative approach. I also appreciate the template-free nature of the approach. The level of detail is appropriate.

Syntax and general paper structure:

1. I would delete the last sentence of the abstract, the one starting with "We hope ...". It is more for the "Conclusions".

Response: Thanks for your suggestions, we move that sentence to the last paragraph of the conclusion in Section 4.5.

2. Introduction: line 20 -22, there is a reference missing (Schwaller et al. 2020, <https://pubs.rsc.org/en/content/articlelanding/2020/sc/c9sc05704h>)

Response: we added this reference, thanks for your kind reminder.

3. Maybe a short summary of the sections in the paper and what they contain should be added at the end of the "Introduction".

Response: we added this to the end of the introduction section.

4. Line 128 specify better equation (2) and what is 'w'.

Response: w is one SMILES token from product or reactants, we added this in the revised version.

5. The data section, line 246, should be moved somewhere else, like in 'Methods'.

Response: thanks for this suggestion, we adjusted the section order, and details are moved to Supplementary Methods.

6. Figure 6, specify "products with CORRECT substructures" and "Products with ALL CORRECT substructures" in the legend.

Response: we fixed this error, thanks for remind us.

7. The section from line 341 on should be the first chapter in "Results".

Response: we adjusted the section order in the revised version.

8. There is no "Conclusion" section which makes it hard for the reader to understand the main findings and impact for the work (you need to read between the lines).

Response: we add discussions and conclusions in Section 4

Work results:

9. How is the validation set used? Hyperparameters search or just checkpoint selection? Should be more specific for both the "Reaction retrieval model" and the "Substructure-level sequence-to-sequence learning" on how the validation is performed.

Response: we add details regarding model settings and how model validation was carried out in the Supplementary Methods section.

10. In paragraph starting from line 141: Suppose You count more than 5 occurrences for a chain of 4 carbon atoms and then you also count more than 5 occurrences for the SAME CHAIN with one atom more (ex. oxygen). You keep both substructures or just the smaller/bigger one? It would be nice to be more specific on this.

Response: The extraction algorithm first uses fingerprints to determine if an atom should be included in the substructure. Once the atoms of substructures are determined, all bonds between those atoms in the original molecule should be retained in the substructure. Taking the example, the 4 carbon atoms and the oxygen should be included in the substructure. We made this clear in the revised version.

11. Line 143-145. It should be shown how you observe that this assumption is reasonable, with statistics.

Response: the assumption that the retrieved candidates share a common substructure with the golden reactants requires that the candidates be structurally similar to the golden reactants. The training objective of dual encoder ensures that the SMILES of retrieved candidates are similar to the golden reactants. Thus, the assumption will be correct if “similar SMILES” is approximately equal to “similar structures”. With common fingerprints calculated among retrieved candidates and the given product, we could further out filter those “similar SMILES” that correspond to different structures.

The assumption is verified in the result section about discussing substructures. We extracted substructures for 81.9% product molecules on the training data, and on the test data, we extracted substructures for 82.2% products with an accuracy of 90.2%. The accuracy and coverage of common substructures indicate that the assumption is reasonable for most cases.

12. Line 186-188. Is there a reason for the choice of representation with charge-neutral molecules instead of radicals?

Response: We initially considered using free radicals as an option to represent the substructures after bond breakage. However, as free radicals, when shown in the figure with a dot, is confusing to a chemist, because free radical reactions refer to a special chemistry mechanism, we opted not using that representation.

13. Line 188-190: Where and how is the single/double/triple bond information stored if the bond is cleaved?

Response: we stored the bond information as properties in the virtually labeled atoms (isotopically labeled atoms in the initial version) using the SetProp API

(<https://www.rdkit.org/docs/source/rdkit.Chem.rdchem.html#rdkit.Chem.rdchem.Atom.SetProp>).

When merging predicted fragments with the substructures, we used the property values of the labeled atoms to build the predicted reactant molecules. The property could also be easily encoded using the extended SMILES format with the MolToCXSmiles API

(<https://www.rdkit.org/docs/source/rdkit.Chem.rdmolfiles.html#rdkit.Chem.rdmolfiles.MolToCXSmiles>).

14. Line 198-207: I disagree on this. I like this augmentation approach, but for the training set only the golden reactants should be used, for example using different radii for the substructure. If all the retrieved reactants are used, considering that some bring incorrect substructures you just introduce more noise in the training. The reaction retrieval model should be used only at test time.

Response: we only used the correct substructures that exists in the golden reactants as the training data, incorrect substructures were filtered out if they were not contained in the golden reactants. These augmented substructures could be considered as if they were extracted from the golden reactants. We explained this in the revised version.

I would like to further explain this with the demo case at

https://github.com/fangleigit/demos/blob/master/demo_19.ipynb.

The above figure shows the input product molecule and the golden reactants. Below are the retrieved candidate reactants (we only show the top 5 here).

After extracting substructures from each candidate, we obtained the results as follows. Here we only show two substructures, and for each substructure, we highlight where it is in the product, the golden reactants, and the candidate(s).

On the training data, incorrect substructures could be easily filtered out by a comparison to the golden reactants, which means that we considered these substructures as if they were extracted from the golden reactants.

Taking Substructure_0 and Substructure_1 as examples, they are slightly different. On the training data, product molecules will be represented multiple times with these correct substructures. In this way, the trained model would be less sensitive to the extracted substructures.

Note that on the training data, we used the correct substructures only, while during inference, we used all extracted substructures. This ensures that the substructures are obtained in a similar way for model training and inference. The performance might drop if we train the model with substructures extracted under different radii from golden reactants and make predictions with substructures extracted from the retrieved candidates.

15. Line 238-243: Not sure I understand why you need a ranking model. Can you just use some “majority” ranking where the top-1 prediction is the one more frequent among the predictions from different substructures from the same target? (After reconstructing the reactants of course).

Response: Top-1 prediction can probably be performed this way. We needed a more rigorous method for other top ranked results. Existing transformer-based approaches use a score function with hyperparameters to rank the predictions. The hyperparameter is tuned on the test set or is empirically set. We added details for this in Supplementary Methods and Supplementary Results.

16. Line 252-254: Is this on top of “substructure augmentation”? So, for each product, for each of its substructure’s representations You add 2 randomized SMILES (ex. If I have 5 different valid substructures for a product -> 15 representations for the same product in the training set?).

Response: yes, this is on top of substructure augmentation. On the test data, the average number of unique substructures is 4.9, thus, the product molecule is represented approximately 15 times. We added more details about this in the Supplementary Methods.

17. Line 277: see previous comment: for training substructures should be extracted only from the golden reactants.

Response: please see our response to that previous question.

18. From line 341 on: are in Table 2 the top-x accuracies from the “substructure-found” test set combined with the “no-substructure-found” test set? They should be split as well.

Response: we thank the reviewer for the suggestion and added the results for products with extracted substructures in the table accordingly.

19. What is the percentage on invalid SMILES prediction compared to other models?

Response: because we only needed to predict the fragments, the length of SMILES is significantly reduced (the average number of heavy atoms in the product, substructures, and golden reactants are 26.3, 12.1, and 30.0, respectively). The table below shows the error rate compared to Augmented Transformer and Graph2SMILES. The results also show that with the length of the output sequence reduced, the possibility of generating invalid SMILES was also reduced.

rank	Graph2SMILES	Augmented Transformer	Our approach
1	1.61%	0.20%	0.11%
2	10.24%	0.81%	0.38%
3	15.85%	1.43%	0.64%
4	19.63%	1.90%	1.02%
5	23.15%	2.29%	1.48%
6	25.60%	2.72%	1.85%
7	26.53%	3.26%	2.09%
8	28.06%	4.19%	2.56%
9	29.26%	6.09%	3.12%
10	30.54%	10.01%	4.65%

Table: Error rate of SMILES

REVIEWERS' COMMENTS

Reviewer #1 (Remarks to the Author):

I have re-read the whole manuscript, along with the authors' reply to my concerns. I am happy the authors agreed, that the manuscript in the previous form was "promising too much", especially for chemistry-oriented readers. In its current form, the scope of the manuscript is better defined, the captions of the Figures are helpful and there are much less misleading statements. I have only minor comments, which in my opinion may help the authors to finalize the manuscript.

There are some technical details that are not clear to me. I might have missed the information, what was the ratio in train/valid split, and were the sets the same for all three modules?

In the description of the reaction retrieval module, I guess 'd' (line 132) is the dimension of the space? And how much was it? Also, capital X and Y in equation 2 should be in "italics", as under the sum sign?

In line 122 - was the [BOS] token added to tokenized smiles (as written), or to smiles and then it was tokenized (as in the Figure)? Also, you introduce E_{pro}, and E_{rea}, which you actually do not use.

What was the final batch size?

I do not understand, what is "one side of product" and "one side of reactant" in lines 135-136. I guess it is a score of "coverage" of one product versus all reactants or vice versa. Still, how it's written is rather vague for me.

Actually, the example in Fig. 1 is slightly misleading, as the "Fragment" remains in the final substrate. I understand that this is a behavior of the model slightly divergent from chemical intuition, which does not cause the model to fail. But I think that the authors may "warn" the reader early, that it is expected, and it is a result of defining "preserved substructures" as the parts of the structures present in ALL similar reaction substrates.

Line 186 - it should be product instead of query.

Line 255 - in SMILES, there is a special character marking reaction - it is ">>" (substrate1.substrates>>product).

I also appreciate the authors' attempt to compare the model to some benchmark model. My previous concern stems from the fact, that Transformer applied to SMILES can produce very creative results, however, not always chemically correct. In fact, it is good to combine transformer (generative model) with some discriminating model, which filters out potentially wrong reactions. One such problem is, that transformer can combine a few reactions into one, which is not chemically

acceptable. This effect is, however, not visible if you calculate only TopK - so you concentrate only on the positive reactions.

Also, to comment on the one-pot reactions - actually, if you look into patents themselves, you would notice that there are quite a few examples, where these are multistep reactions, which are then combined into one. In particular, in the example of amide coupling, it is really common to first protect one reacting group, conduct the reaction and then deprotection - all in one step. Therefore it is highly interesting, that your model could efficiently discriminate the correctly reacting group.

To sum up, I think that the manuscript is much clearer now than in the previous form and I appreciate the work the authors have already done. I left some comments, which are not crucial, but can addressing them in the manuscript could make it easier to read in my opinion.

Reviewer #2 (Remarks to the Author):

The authors have addressed all my concerns and I am satisfied with their answers. I just believe that the English language still needs some polishing, but in general I am satisfied with the modifications to the manuscript.

Reviewer #1 (Remarks to the Author):

I have re-read the whole manuscript, along with the authors' reply to my concerns. I am happy the authors agreed, that the manuscript in the previous form was "promising too much", especially for chemistry-oriented readers. In its current form, the scope of the manuscript is better defined, the captions of the Figures are helpful and there are much less misleading statements. I have only minor comments, which in my opinion may help the authors to finalize the manuscript.

There are some technical details that are not clear to me. I might have missed the information, what was the ratio in train/valid split, and were the sets the same for all three modules?

The ratio of train/valid/test is 80%/10%/10%, which were moved from the Supplementary Methods to the main article. The train/valid split was same for our model and the vanilla AugmentedTransformer model. The training/val data for the ranking model is collected from the valid data only. This was also moved from the Supplementary Methods to the manuscript.

In the description of the reaction retrieval module, I guess 'd' (line 132) is the dimension of the space?

And how much was it?

Yes, it is the "Hidden size" in the Table about parameter settings (Supplementary Table 3 in our previous submission, and now Table 2 in the revised version), which was set to 256 in our experiments. We added text under the dual encoder figure (Figure 6) to make it clearer.

Also, capital X and Y in equation 2 should be in "italics", as under the sum sign?

The capital X and Y represent the set of tokens in product SMILES and reactants SMILES, respectively. We keep them as they were because the symbols are already in italics.

In line 122 - was the [BOS] token added to tokenized smiles (as written), or to smiles and then it was tokenized (as in the Figure)?

[BOS] token will be added to the tokenized SMILES. And in the figure the SMILES were also tokenized, we made this clear in the figure.

Actually, the final sequence would be the same in both scenarios. We follow <https://github.com/pschwillr/MolecularTransformer#pre-processing> in tokenizing the SMILES, and [XX] will be tokenized as a single token, e.g., the virtually labeled atoms will also be tokenized as single tokens.

Also, you introduce E_{pro}, and E_{rea}, which you actually do not use. What was the final batch size?

E_{pro} and E_{rea} are the representations for the product and reactant, respectively. E_{pro} and E_{rea} are used to calculate the score for one entry. We added text about how E_{pro} and E_{rea} are related to X and Y in the manuscript. The batch size to train the dual encoder is 4096, which we moved from the supplementary method to the main article.

I do not understand, what is "one side of product" and "one side of reactant" in lines 135-136. I guess it is a score of "coverage" of one product versus all reactants or vice versa. Still, how it's written is rather vague for me.

Yes, we polished the writing of the whole paper. Thank you for pointing this out.

Actually, the example in Fig. 1 is slightly misleading, as the "Fragment" remains in the final substrate. I understand that this is a behavior of the model slightly divergent from chemical intuition, which does not cause the model to fail. But I think that the authors may "warn" the reader early, that it is expected, and it is a result of defining "preserved substructures" as the parts of the structures present in ALL similar reaction substrates.

Thanks for this remainder, which would make our paper clearer. We added this warning in the manuscript.

Line 186 - it should be product instead of query.

We changed the query to product in the revised version.

Line 255 - in SMILES, there is a special character marking reaction - it is ">>" (substrate1.substrates>>product).

The special character "|" was introduced as a separator between the substructure and fragments in the product. In order not to mislead readers, we changed

"|" is a special character indicating the start of product fragments SMILES.

to

"|" is a special character indicating the start of fragments SMILES in product.

I also appreciate the authors' attempt to compare the model to some benchmark model. My previous concern stems from the fact, that Transformer applied to SMILES can produce very creative results, however, not always chemically correct. In fact, it is good to combine transformer (generative model) with some discriminating model, which filters out potentially wrong reactions. One such problem is, that transformer can combine a few reactions into one, which is not chemically acceptable. This effect is,

however, not visible if you calculate only TopK - so you concentrate only on the positive reactions. We did not notice that multiple reactions are combined into one, thanks for your clarification on this, which deepen our understanding of this. We will investigate the patents in our future work.

Also, to comment on the one-pot reactions - actually, if you look into patents themselves, you would notice that there are quite a few examples, where these are multistep reactions, which are then combined into one. In particular, in the example of amide coupling, it is really common to first protect one reacting group, conduct the reaction and then deprotection - all in one step. Therefore it is highly interesting, that your model could efficiently discriminate the correctly reacting group.

We didn't investigate the patents, we appreciate your suggestions, and will go through details of the patents in our future work. Retrosynthesis in "on-pot reactions" could be one interesting research topic in our future studies.

To sum up, I think that the manuscript is much clearer now than in the previous form and I appreciate the work the authors have already done. I left some comments, which are not crucial, but can addressing them in the manuscript could make it easier to read in my opinion.

Reviewer #3 (Remarks to the Author):

The authors have addressed all my concerns and I am satisfied with their answers. I just believe that the English language still needs some polishing, but in general I am satisfied with the modifications to the manuscript.

We have our paper polished by native speakers.